cognition

accumulator model, animal cognition, inhibition of return, magnitude, spatial frequency

**Author for correspondence:**
Cwyn Solvi
e-mail: c.solvi@qmul.ac.uk

†Authors contributed equally to this work.

# Non-numerical strategies used by bees to solve numerical cognition tasks

HaDi MaBouDi[1,†], Andrew B. Barron[1,2,†], Sun Li[3], Maria Honkanen[4], Olli J. Loukola[4], Fei Peng[3], Wenfeng Li[5], James A. R. Marshall[1], Alex Cope[1], Eleni Vasilaki[1] and Cwyn Solvi[2,6]

[1]Department of Computer Science, University of Sheffield, Sheffield S1 4DP, UK
[2]Department of Biological Sciences, Macquarie University, North Ryde, New South Wales 2109, Australia
[3]Department of Psychology, School of Public Health, Southern Medical University, Guangzhou, People's Republic of China
[4]Ecology and Genetics Research Unit, University of Oulu, Oulu, Finland
[5]Guangdong Key Laboratory of Animal Conservation and Resource Utilization, Guangdong Public Laboratory of Wild Animal Conservation and Utilization, Institute of Zoology, Guangdong Academy of Science, Guangzhou, People's Republic of China
[6]School of Biological and Chemical Sciences, Queen Mary University of London, London E1 4NS, UK

HDM, 0000-0002-7612-6465; ABB, 0000-0002-8135-6628; MH, 0000-0002-2729-5588; OJL, 0000-0002-9094-2004; FP, 0000-0002-1637-5611; CS, 0000-0003-2517-6179

We examined how bees solve a visual discrimination task with stimuli commonly used in numerical cognition studies. Bees performed well on the task, but additional tests showed that they had learned continuous (non-numerical) cues. A network model using biologically plausible visual feature filtering and a simple associative rule was capable of learning the task using only continuous cues inherent in the training stimuli, with no numerical processing. This model was also able to reproduce behaviours that have been considered in other studies indicative of numerical cognition. Our results support the idea that a sense of magnitude may be more primitive and basic than a sense of number. Our findings highlight how problematic inadvertent continuous cues can be for studies of numerical cognition. This remains a deep issue within the field that requires increased vigilance and cleverness from the experimenter. We suggest ways of better assessing numerical cognition in non-speaking animals, including assessing the use of all alternative cues in one test, using cross-modal cues, analysing behavioural responses to detect underlying strategies, and finding the neural substrate.

## 1. Introduction

Mapping specific cognitive capacities to the behaviour of any animal is rarely straightforward. The difficulty is that animals may not be solving the task the way we think they are. One example of this is in our own recent work where we had bees discriminate different shapes based on relative size [1]. Bees' performance increased over training to well above chance, and in the unrewarded test they seemed to have learned to discriminate shapes based on relative size. However, analysis of first and sequential choices during training bouts and tests revealed that the bees actually switched to a simpler strategy in the middle of training: win-stay/lose-switch. These results, along with other works suggesting animals are able to solve tasks in unexpected ways (e.g. [2–7]), prompted us to look deeper into the strategies of animals in numerical cognition tasks.

Numerical cognition has been claimed in a large number of animal species (e.g. [8–40]), suggesting that a sense of number is widespread (for reviews see [41–43]). By far, the most common method for testing numerical cognition in non-verbal animals is to have subjects discriminate two-dimensional visual displays with differing numbers of shapes (figure 1; [8–40] all used this

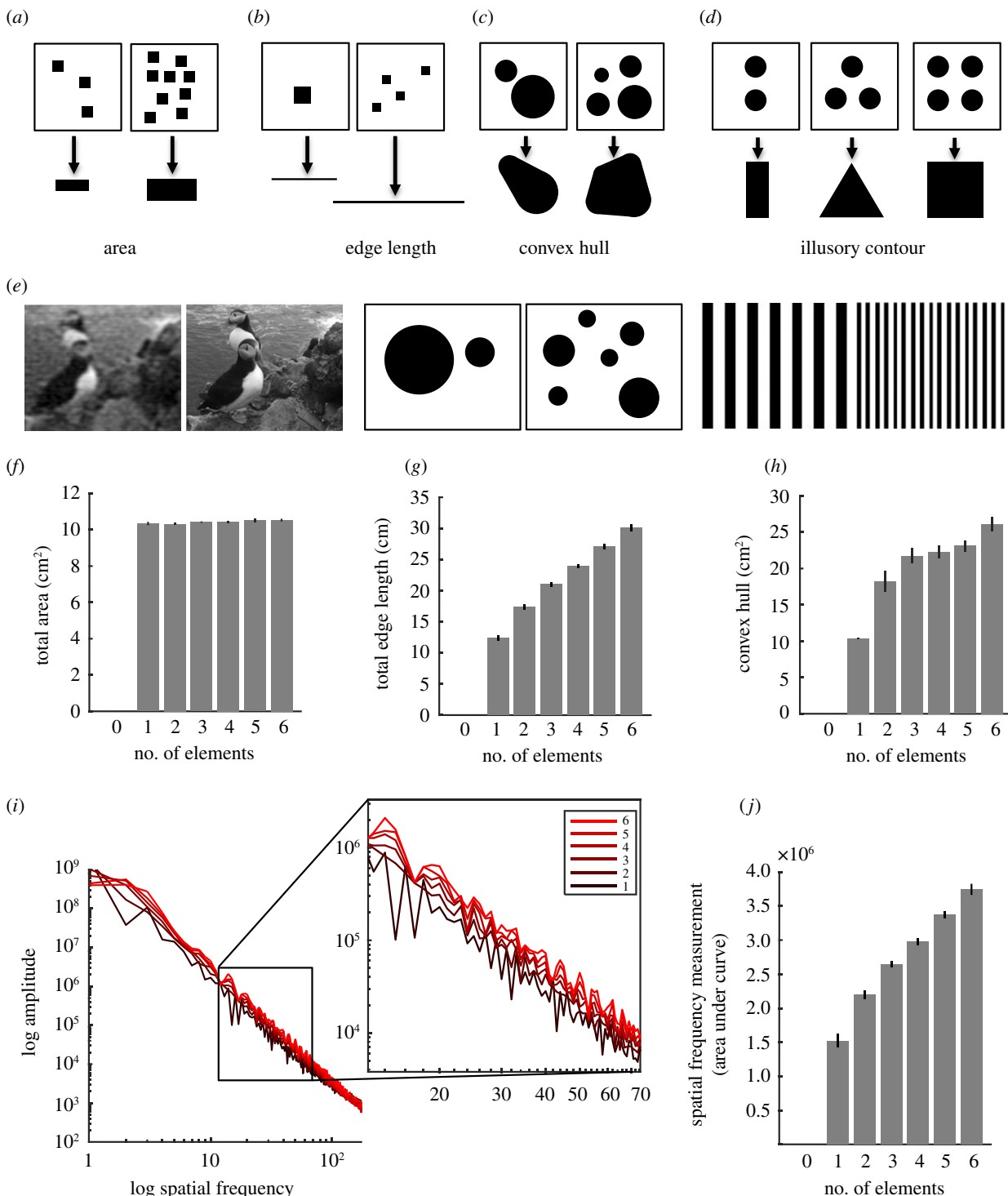

**Figure 1.** Number of elements naturally covaries with non-numerical cues. (*a–d*) Examples of two-dimensional stimuli used in numerical cognition studies and how different continuous cues normally covary with numerosity. Note that illusory contour does not covary with numerosity but can still be learned and used to solve numerical cognition tasks, especially with lower numbers of elements. (*e*) Spatial frequency (the amount of alternating dark and light regions in a given area) also normally covaries with numerosity. The more changes from black to white across an image in all directions, the greater spatial frequency. The right images of each pair in (*e*) all have higher spatial frequency than the left images. (*f–h*) For all stimuli in [28], from which our stimulus set was borrowed, area (amount of total black) was kept constant (*f*), but edge length (total boundary length; *g*) and convex hull (the minimum convex region covering all elements; *h*) covaried with numerosity. (*i,j*) Spatial frequency is calculated by obtaining a power spectrum (Methods) and measuring the area under the power spectrum's curve. The power spectrum plots (*i* and zoomed-in inset) for all stimuli in [28], from which our stimulus set was borrowed, averaged for each number of elements from one to six, shows that spatial frequency increases with numerosity (*j*). Note that for all covarying continuous cues, a zero-set stimulus will have zero measurement and thereby be placed naturally at the lower end of the spectrum for each of these non-numerical cues. (Online version in colour.)

design). As pointed out by others (e.g. [44,45]), in these types of designs, continuous (non-numerical) cues often unavoidably covary with numerosity. These include size and shape of elements, area (total amount of colour), edge length (total boundary length of elements), convex hull (the minimum convex region covering all elements), spatial frequency (the amount of alternating dark and light regions), and illusory contour (the basic shape that outlines all

elements). In figure 1, we further describe these cues and their natural covariation with number (figure 1a–e). This covariation makes it difficult to know whether animals actually used any sense of number to solve their tasks.

The issue of non-numerical strategies within numerical cognition studies has been highlighted by others [44–47]. It was established decades ago that cells within the visual system respond to various continuous visual features [46,48,49] and it has long been known that continuous features can be reliable discrimination cues, even for bees [50–53]. Further, several works show that animals use non-numerical cues to solve numeric-based tasks when not controlled for, e.g. size of elements [54], total area [55] and convex hull [56], and even when they are controlled (e.g. [57]; see Discussion).

Most studies investigating numerical cognition attempt to control for at least one non-numerical cue. Several works have made valiant efforts to control for most continuous cues (e.g. [58,59]). However, we have found no studies that tested for all continuous variables. It seems clear that animals are solving these tasks, but the question we attempt to address here is how they might be solving them. We set out to determine how honeybees solve a numeric-based task using stimuli common among numerical cognition studies.

## 2. Material and methods

### (a) Subjects

Honeybees (*Apis mellifera*) used in the experiment were maintained at the University of Oulu (Oulu, Finland) and at Guangdong Institute of Applied Biological Resources (Guangzhou, China) in September and November 2019, respectively. Prior to training, honeybees fed ad libitum from a gravity feeder providing 30% sucrose solution. Each focal honeybee was first lured to visit the experimental setup by allowing her to drink and walk onto a cotton bud soaked in 50% sucrose solution and then transferring the bee to the setup. Each forager that returned to the setup on her own was marked on her thorax with a coloured dot for identification.

### (b) Experimental setup and procedure

The setup consisted of a 50 × 50 cm acrylic sheet. Stimuli were 6 × 6 cm white displays (laminated sheets of paper) with between one and four black shapes (squares, diamonds or circles). The stimuli, identical to those used in [28], were presented vertically with a landing platform attached just below the displays. Stimuli were randomly allocated for each bee and changed when the bee returned to the hive to offload sucrose, prior to her returning to the setup. The spatial arrangement of stimuli could be randomly changed, thus excluding position orientation cues. The background acrylic sheet and landing platforms were grey coloured. The acrylic background sheet, platforms and displays were washed with 70% ethanol between all visits to exclude the use of olfactory cues. Two shapes were used in training, and the third shape was used for testing. During training, honeybees found either a 10 µl droplet of 50% sucrose solution or 60 mM quinine hemisulphate solution, for correct and incorrect choices, respectively. Each trial, four stimuli (two identical correct; two identical incorrect) were presented simultaneously on the acrylic sheet. Stimuli positions were changed after each choice to new random positions. A choice was defined as any time a honeybee landed on a platform and touched the solution (sucrose or quinine) with their proboscis, leg or antenna.

One group of bees (n = 10) was trained to associate stimuli consisting of more elements with reward, and a second group of bees (n = 10) was trained to associate stimuli consisting of fewer elements with reward. The choices of individual bees during training were recorded until a criterion of greater than or equal to 80% for any 10 consecutive choices was reached (after a minimum of 20 choices). Once an individual bee reached criterion, she was presented with a learning test followed by two additional control tests. Bees reached criterion on average in 41 ± 8 choices. Each test lasted 2 min and all choices made were recorded as the dependent variable for statistical analyses. During all tests, 10 µl of unrewarding water was placed on each platform. Between tests, bees received two reinforced refresher trials (with the same stimuli used in training) to maintain motivation. For the learning test, bees were presented with the shape that they had not been trained on—the purpose being to test whether bees learned to solve the task. The two control tests examined whether honeybees used the number of elements or continuous visual cues. The first control test (equal/incongruent test) had two pairs of stimuli, each with two elements, but one pair with higher edge length, convex hull and spatial frequency. The second control test (incongruent/opposite test) also had two pairs of stimuli, one pair with three elements and the other with two elements but with higher edge length, convex hull and spatial frequency. In all tests, the total black surface area was the same across all stimuli. Experiments were performed by three different groups of individuals (M.H. and O.J.L., S.L. and C.S.) to help independently verify the results.

### (c) Statistical analyses

R 3.6.1 with library 'lme4' was used to perform all generalized linear mixed-effect models (GLMM) with binomial distribution and logit function. For the GLMM evaluating the results of the tests, country and rule (more-than/less-than) were considered as fixed factors and bee identity as a random effect (electronic supplementary material, table S1). Because country and rule had no effect on performance, we display data as the mean ± s.e.m. of all bees' data. We then removed country and rule in a second GLMM (electronic supplementary material, table S2). Our second model ranked better than the first on the grounds of Akaike's information criterion [60] adjusted for small sample sizes (AICc), and therefore we present data from this second model in the main text. For analyses of all test videos, a blind protocol was employed, in that each video filename was coded so that the experimenter doing the analysis was blind to the training of each bee.

To calculate the spatial frequency of the training and test stimuli, a two-dimensional Fourier transform on each image was performed, followed by a power spectrum calculation as the square amplitude of the Fourier transform and averaged over orientation [61]. The actual power over all frequencies was then measured by calculating the area under the curve of the radially averaged power spectrum. Calculations for area, spatial frequency, convex hull and edge length were done in MATLAB 2018b (MathWorks, Mass., USA). Statistical analyses for the model results were also performed in MATLAB 2018b.

### (d) Neural network model

Our model uses spatial frequency encoding that is supported by bees' ability to discriminate visual patterns based on spatial frequency [50,51] and observed neurons in the visual lobe of insects that provide a mechanism of frequency coding [62,63]. Analogous to the spatial frequency coding in primates [64,65], bees may use Gabor-like filters in their visual lobe to extract spatial frequency information from visual stimuli [66]. For our model, the stimulus, $s$, is encoded by the activity of a population of neurons with different preferred spatial frequency that possess similar response profiles. The evoked spiking activities of the

seven sensory neurons were simulated by fixed Gaussian tuning curves spanning spatial frequencies of the input from zero to six as

$$g_i\,(s,\sigma) = R_0 + R_{Max}\,\exp\left[-\frac{1}{2\sigma^2}(s - f_i\,)^2\right] + \aleph(0,\sigma_N),$$

where $R_0 = 50$ spike s$^{-1}$ and $R_{Max} = 200$ spike s$^{-1}$ are the spontaneous and maximum firing rate of the sensory neuron. $\sigma = 2.5$ controls the degree of the selectivity of the sensory neurons to different frequencies around the preferred frequency, $f_i$. Gaussian noise, $\aleph(0,\sigma_N)$ model the randomness of neural activities.

Outputs of all sensory neurons drive a decision neuron through a vector of synaptic weights, $W$, to create the decision neuron's activity in response to the input, as

$$D(s) = F\left(\sum_{k=0}^{6} W_k \cdot g_k(s,\sigma); a,b\right),$$

where $F(x;\,a,\,b) = A_0/(1 + \exp(-a(x-b)))$ is the activation function with the maximum activity at $A_0 = 100$ spike s$^{-1}$. The parameters $a = 0.05$ and $b = 50$ control the sensitivity of the neuron to the input and spontaneous activity of the decision neuron, respectively.

Because we assume that the difference of the decision neuron's responses to the positive ($s_p$) and negative stimuli ($s_n$) must be increased during the training phase, the locally optimal synaptic weights, $W^{opt}$, can be obtained from maximizing the objective function:

$$L = \sum_{t=1}^{m}[D(s_p^t) - D(s_n^t)]\,r^t,$$

where $t$ and $m$ are the index over the paired stimuli and the number of presented stimuli, respectively. Here, $r$ presents the reinforcement signal (VUM-mx1 neuron) that provides modulated feedback whether a stimulus is paired with the reward or punishment ($r = 1$) and $r = 0$ for when no reinforcement signal is presented. The (online) updates of the synaptic weights, $W_i^t$ are calculated by

$$W_i^t = W_i^{t-1} + \eta\,\frac{\partial}{\partial W_i}(D(s_p^t) - D(s_n^t))\,r^t,$$

where $\eta$ is the rate of the weights change. $W_i^{t-1}$ is the updated weight from the iteration $t-1$ (with $W_i^0$ being the initial weight), and

$$\frac{\partial}{\partial W_i}(D(s_p^t) - D(s_n^t)) = g_i(s_p^t,\,\sigma)\,F'\left(\sum_{k=0}^{6} W_k \cdot g_k(s_p^t,\sigma); a,b\right)$$
$$- g_i(s_n^t,\sigma).F'\left(\sum_{k=0}^{6} W_k \cdot g_k(s_n^t,\sigma); a,b\right).$$

Finally, the derivative of the activation function $F$ is obtained as

$$F'(x;\,a\,,b) = \frac{A_0\,a\,\exp(-a(x-b))}{(1 + \exp(-a(x-b)))^2}.$$

After exposing the model to conditioned stimuli in learning paradigms, the behavioural outcomes of the model presented with a pair of the test stimuli were evaluated as a simple subtraction of the decision neuron's responses to both test stimuli.

# 3. Results

## (a) Bees use continuous cues over numerosity in a numerical cognition task

Using the same two-dimensional visual stimulus set as a paradigmatic honeybee study [28], and similar to stimulus sets used for other animals (e.g. [8–40]), we first asked whether honeybees use numerosity to solve a numeric-based discrimination task. In this particular stimulus set, area (total black within each stimulus) is kept constant across all stimuli, and therefore could not be used to solve the task. However, similar to many other numerical cognition studies, edge length (Spearman correlation: rho = 0.93, $p = 1.00 \times 10^{-40}$), convex hull (Spearman correlation: rho = 0.44, $p = 4.88 \times 10^{-6}$) and spatial frequency (Spearman correlation: rho = 0.92, $p = 1.00 \times 10^{-40}$) covaried with number (figure 1f–j). We, therefore, aimed to train bees on this stimulus set, for which they have already been shown to discriminate, and subsequently test bees to determine whether they had used these particular continuous cues or numerosity to solve the task.

We first trained honeybees ($n = 10$) to find rewarding sugar solution on displays with more shapes and an aversive quinine solution on displays with fewer shapes (Methods; figure 2a). Another group of bees ($n = 10$) was trained on the opposite contingency. Once bees reached 80% performance (8 out of 10 consecutive choices correct), they were given an unrewarded learning test. Bees trained on a 'more-than' rule preferred (landed on more often) stimuli containing more elements during the test, whereas bees trained to 'less-than' preferred stimuli with fewer elements. Honeybees showed high performance in the learning test (figure 2b left; GLMM: 95% confidence interval (CI) = 0.75 (0.47 to 1.03), $n = 20$, $p = 1.49 \times 10^{-7}$).

To determine if bees used non-numerical cues, after the learning test and refresher trials (Methods), we tested the same honeybees on an 'equal/incongruent test', where two pairs of unrewarded stimuli contained the same number of elements (figure 2b middle), but differed in edge length, convex hull and spatial frequency (figure 2c–f). If honeybees were using numerosity, they should prefer all displays equally during this test. Conversely, honeybees more often chose stimuli with a higher level of continuous variables if they had been trained to choose stimuli with more elements, and more often chose stimuli with a lower level of continuous variables if they had been trained to choose stimuli with fewer elements (figure 2b middle; GLMM: 95% CI = −0.64 (−0.89 to −0.39), $n = 20$, $p = 6.5 \times 10^{-7}$). This suggests honeybees responded to continuous cues in the stimuli and not the number of elements.

We further tested honeybees on an 'incongruent/opposite test' where the number of elements in each pair of displays differed (figure 2b right) and the continuous cues (edge length, convex hull and spatial frequency) were in the opposite direction to the numerical difference (i.e. higher for two elements than for three elements; figure 2c–f). In this test, honeybees behaved in the reverse manner to which we would expect if they had learned numerosity. Bees that were trained to associate more elements with reward preferred test displays with the higher level of continuous variables but fewer elements. Bees that were

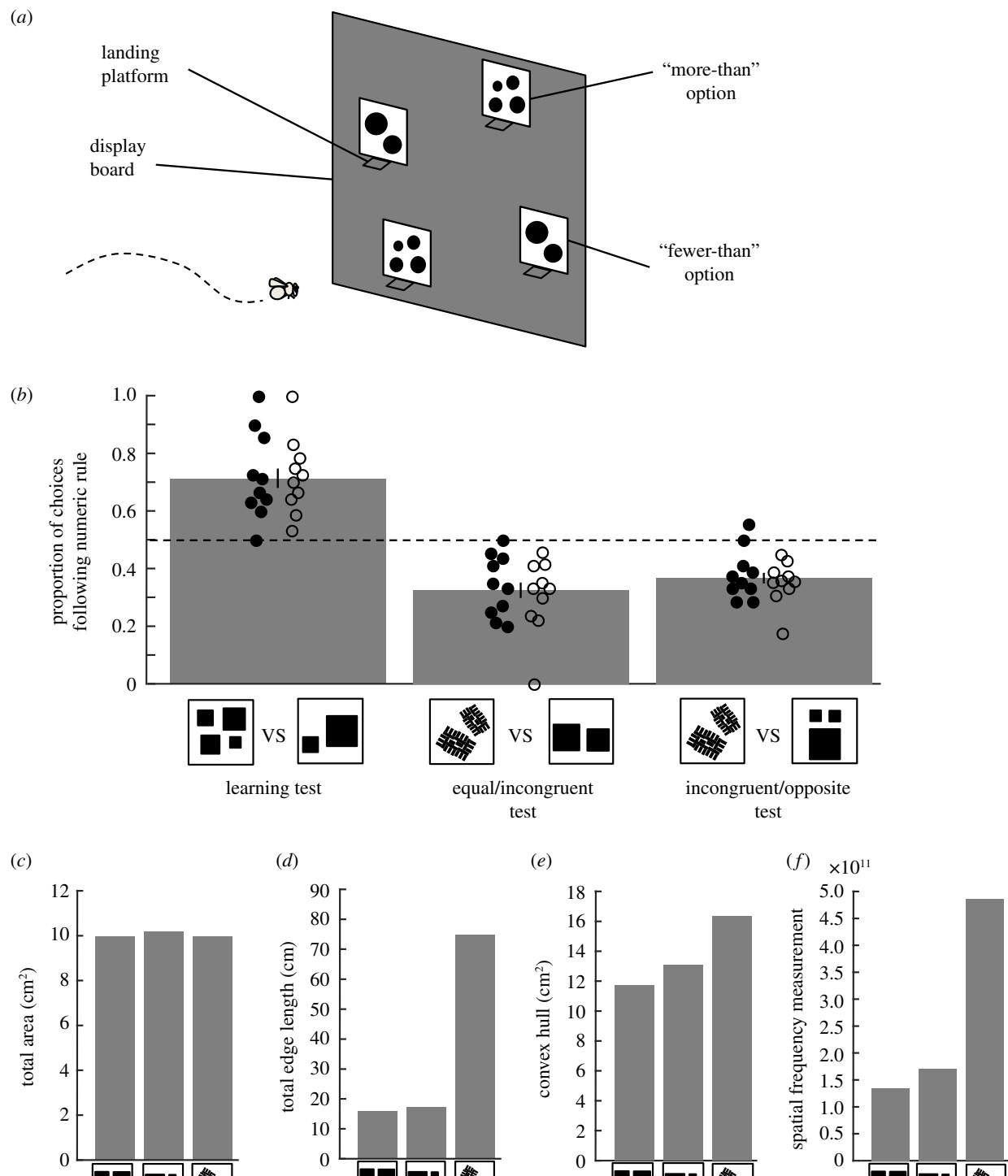

**Figure 2.** Bees can use non-numerical strategies to discriminate numerical stimuli. (*a*) Experiment setup. Honeybees were trained to find 50% sucrose solution at one of two pairs of displays showing different numbers of elements, and aversive quinine solution on the other display pair (Methods). (*b*) Once honeybees reached 80% performance, they were tested using displays with novel shapes. In the learning test, honeybees more often chose stimuli following the numerical rule on which they had been trained (71.3 ± 3.3%; more-than: 70.3 ± 4.7%; less-than: 72.4 ± 4.8%). However, when tested on stimuli that differed in continuous cues but not number of elements (equal/incongruent test; middle bar; 32.5 ± 2.6; more-than: 30.7 ± 4.2%; less-than: 34.2 ± 3.4%) and separately on two pairs of stimuli where numerosity and continuous cues were set in opposition (incongruent/opposite test; right bar; 36.7 ± 1.8; more-than: 35.1 ± 2.4%; less-than: 38.2 ± 2.8%), honeybees chose stimuli based on continuous cues over numerosity. Data shown are combined from the two groups trained with different numerical rules as no difference in performance was found between groups (electronic supplementary material, table S1; Methods). Dotted line = 0.5 chance level. Bars = mean. Vertical lines = s.e.m. Circles = individual bees' data points (filled circles: bees trained to more-than rule; empty circles: bees trained to less-than rule). (*c–f*) Stimuli used in tests with corresponding continuous variable measurements (Methods).

trained to associate fewer elements with reward preferred test displays with the lower level of continuous cues but more elements (figure 2*b* right; GLMM: 95% CI = −0.55 (−0.79 to −0.30), $n = 20$, $p = 1.17 \times 10^{-5}$).

Our results indicate that honeybees use continuous properties to discriminate stimuli with varying numbers of shapes. This caveat may also apply to other numerical cognition studies with honeybees and other animals.

## (b) A neural network model with no reference to numerosity can reproduce behaviours indicative of numerical cognition

Our results beg the question: what explanation is simpler and more plausible: numerical or non-numerical processing? Therefore, how simple is learning continuous variables as an explanation for the behaviour of honeybees? To explore this, we created a simple neural network model containing just nine neurons arranged in three layers (figure 3a) to encode a relational rule (more-than or less-than) based only on one non-numerical cue (Material and methods). Seven sensory neurons encoded spatial frequency in the visual lobe which projected frequency information to the eighth neuron, a single decision neuron. Synaptic weights between the sensory neurons and decision neuron were adjusted according to the activation (by the presentation of stimuli) of the ninth neuron, a reinforcement neuron based on the specific learning rule (more-than or less-than). We chose spatial frequency for simplicity, and because we have yet to find any recent study that controlled/tested for it, but the model could also be applied to other continuous variables.

We trained our model following the methods for several experiments in [28], a recent study that had honeybees discriminate two-dimensional visual cues with different numbers of shapes. We then evaluated the model's choices when presented with test stimuli (see Methods for details and figure 3 for simplification). This simple model was able to reproduce the behaviour of honeybees in numerical cognition tasks, with a very simple computational structure using only non-numerical information. Specifically, the model could transfer a 'more-than' or 'less-than' rule to novel shapes, to stimuli containing a number of elements outside the range trained on and to stimuli with zero elements, and could recognize stimuli with zero elements as the lower end of a continuum (figure 3b–e). Thus, we are able to reproduce behavioural evidence that has been taken in honeybees (and similarly in other animals) as indicative of understanding number with a model in which there is no processing of numerosity.

## 4. Discussion

### (a) General summary

We are not suggesting that all numerical cognition studies are wrong or that no animal has numerical cognition. We show, however, that in a task using a two-dimensional visual display set with differing numbers of shapes, non-numerical cues can be learned, they dominate over numerosity when equal to or set in opposition to numbers of elements, and they can be learned by simple computational systems with no reference to numerosity. Our behavioural and computational results provide a counter example against the assumption that two-dimensional visual stimuli with different numbers of shapes are processed by honeybees as discrete numerical elements. Our findings suggest that an alternative non-numerical explanation exists for studies using similar methods in honeybees. If other animals are sensitive to any available continuous cues, then an alternative non-numerical explanation exists for those results as well. This is vital information if we truly want to know how any animal solves the numerical problems they face in their own ecological niches.

### (b) The depth of the issue

It is very difficult to control for all continuous visual cues [45,67]. By controlling one parameter, another will necessarily covary with numerosity. Even varying parameters randomly during training is not enough to solve the issue. Leibovich & Henik [57] trained adult humans on visual stimuli of differing numbers of dots where continuous cues were minimally correlated or uncorrelated with numerosity. Despite this, they found that in a regression analysis, half of the behavioural variance could be explained by the irrelevant continuous cues [57]. Presenting stimuli separately/sequentially may make the task more difficult (e.g. [68,69]). However, animals may store, in working memory, an accumulation of neural responses to continuous variable changes as they pass/observe stimuli, without reference to numerosity [70–73].

It will also not suffice to test for continuous cues separately because animals may learn multiple redundant cues and use those available when others are not [74–79]. Testing all continuous variables (that cannot be kept constant across stimuli for the entirety of training) and numerosity within one test can help determine if continuous variables have been learned. In one of our recent works, examining how bumblebees solved a numeric-based task, we assessed the use of continuous cues within one unrewarded test [80]. Here, bees were shown 10 stimuli simultaneously during one unrewarded test, each with different numbers of elements and levels of continuous cues. We chose the characteristics of different stimuli so that the bees' choices for some over others would reveal whether or not they had learned and used specific continuous cues to solve the task. For example, two displays both contained the same number of elements, but the elements in one of the displays had a greater edge length. Bees chose these two displays equally in the test, suggesting they did not use edge length. However, if they had performed well on the test (i.e. more often chose stimuli based on the numerosity rule they had been trained) but had chosen one of these two stimuli significantly more than the other, this would suggest bees had learned and used edge length instead of numerosity. We provided pairs of stimuli that varied in this way for edge length, area, convex hull, spatial frequency and illusionary contour. We must keep in mind, as pointed out above, that even when this type of design suggests continuous cues were not used, as it had in our work, other strategies could still be used. Although bees' behaviour [80] indicated some form of counting, the bumblebees could have used working spatial memory to avoid recently visited shapes (cf. 'inhibition of return' [81,82]). Therefore, it is possible that bees discriminated stimuli based on the duration of time taken to scan all shapes within a display, or perhaps by an accumulator mechanism responding to visual changes as they scanned past each shape [70]. Either of these possible strategies do not require a true sense of number.

### (c) Ways forward

How then can we address this natural, deep-seated issue? We propose that the method of assessing all continuous cues in one unrewarded test, in conjunction with varying all continuous cues during training, be set as a minimum when investigating numerical cognition in animals. However,

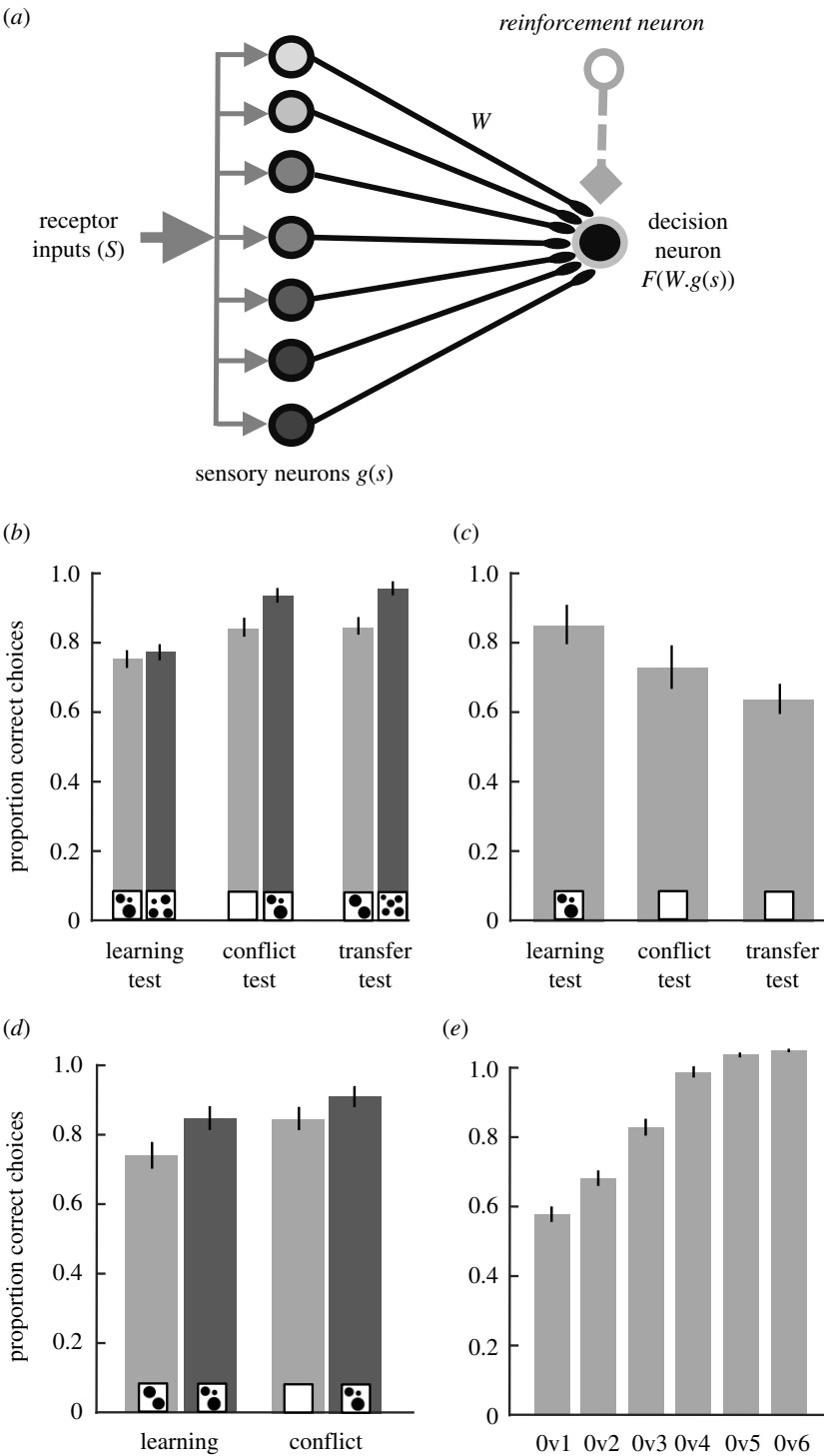

**Figure 3.** A simple computational model using only non-numerical cues reproduces honeybees' performance in a numerosity tasks. (*a*) The model uses seven sensory neurons that are activated by the output of visual receptors. Each sensory neuron responds to multiple levels of a single continuous cue with different sensitivities. Firing of each sensory neuron is specific and selective to the preference level modelled by a Gaussian tuning curve. Information from all sensory neurons converges at a single decision neuron. Synaptic connectivity ($W$) between sensory neurons and the decision neuron are modified by an associative learning rule for encoding appetitive and aversive valences. Performance of the model is evaluated by a simple subtraction of the responses of the decision neuron to the test stimuli. (*b*–*e*) Our model is able to reproduce behaviours claimed to be indicative of numerical cognition [28], without any reference to numerosity. (*b*) Our model can transfer a 'more/less-than' rule to novel shapes in a 'conflict test' examining preference for zero numerosity (Wilcoxon signed-rank test, $z$-value $> 6.22$ and $p < 3.50 \times 10^{-9}$) and a 'transfer test' using displays with more shapes than in training (Wilcoxon signed-rank test, $z$-value $> 7.99$ and $p < 3.17 \times 10^{-14}$). Compare to fig. 1c in [28]. (*c*) Our model can transfer a 'more/less-than' rule to stimuli containing a number of elements outside the training stimuli range, in a learning test (Wilcoxon signed-rank test, $z$-value $= 3.89$ and $p = 9.98 \times 10^{-5}$), conflict test ($z$-value $= 3.23$ and $p = 0.0012$) and transfer test ($z$-value $= 2.40$ and $p = 0.016$). Compare to fig. 1d in [28]. (*d*) Our model can transfer a 'more/less-than' rule to novel pairs of stimuli, including stimuli with zero elements, in a learning test (Wilcoxon signed-rank test, $z$-values $> 5.27$ and $p < 1.35 \times 10^{-6}$), and conflict test (Wilcoxon signed-rank test, $z$-values $> 5.51$ and $p < 3.49 \times 10^{-7}$). Compare to the electronic supplementary material, fig. S4 in [28]. (*e*) Our model can recognize stimuli with zero elements as the lower end of a continuum (Wilcoxon signed-rank test for comparing each pair with the chance level 50%, $z$-values $> 2.24$ and $p < 0.024$; Kruskal–Wallis test, $\chi^2_{299} = 183.94$ and $p = 7.71 \times 10^{-37}$. Compare to fig. 2b in [28]. Light grey, less-than; dark grey, more-than; insets, test stimuli; bars, mean; vertical lines, s.e.m. calculated from the firing rate of the decision neuron for 50 different initial parameters that simulated 50 different model bees.

as mentioned above we need to still keep in mind other potential non-numerical strategies.

Most numerical cognition studies use visual stimuli. Stimuli in other modalities come with their own set of issues regarding continuous variables. For example, number of individuals covaries with the overall complexity of their chemical/olfactory cues, and with the total volume and complexity of vocal calls. However, combining modalities does offer some promising avenues for investigation. One of the strongest pieces of evidence for numerical cognition is the ability to transfer across modalities, which seems to prevent the use of continuous cues because the only similarity across modalities should be numerosity. A nice example of this was shown in monkeys where they were able to match the sum of randomly ordered sequentially presented shapes and tones to a visual array with the same number of squares [83]. This kind of cross-modal generalization design would certainly strengthen arguments for numerical cognition in other animals.

Videos of animals solving numerical cognition tasks can help determine how animals are solving those tasks (cf. [1,2]). Automated approaches combining machine vision and learning with computational behavioural analyses have the ability to discover behavioural features that humans cannot (cf. [84,85]). For example, by measuring the inspection behaviour (e.g. gaze, body direction, movement) of an animal towards different numerical stimuli and comparing across different decisions (choose/reject) and different outcomes (correct/incorrect), underlying strategies may become apparent.

Ultimately, however, we must also establish the underlying neural mechanisms to truly know which cues and strategies an animal used to solve a numeric-based task. This will provide vital information for how numerical cognition may have evolved, and how processing of numerosity compares between animals [86,87].

Data accessibility. The data supporting the findings of this study (figure 1f–j, figure 2b and figure 3b–e), the code necessary for the model, and the code for measuring the continuous visual features of the stimuli are available in the public repository figshare at https://figshare.com/s/21c5753e31f51ece5f1c.

Authors' contributions. H.M., A.B.B and C.S. conceptualized the project and designed experiments. S.L., M.H., O.J.L. and C.S. conducted experiments with the help of F.P. and W.L. C.S. analysed the behavioural data with help from H.M. and F.P. H.M. analysed visual stimuli and created and analysed the model with helpful comments from A.B.B., J.A.R.M., A.C. and E.V. H.M., A.B.B. and C.S. wrote the paper with helpful comments from the other authors.

Competing interests. All authors declare no conflict of interest.

Funding. This study was supported by the EPSRC programme grant Brains-on-Board (EP/P006094/1) awarded to J.A.R.M. and E.V. A.B.B. and J.A.R.M. were supported by a Leverhulme visiting professorship. A.B.B. and C.S. were supported by the Templeton World Charity Foundation project number TWCF0266. F.P. was supported by the National Natural Science Foundation of China (Project no. 31700988). O.J.L. was supported by The Academy of Finland, grant no. 309995.

Acknowledgements. We thank Yonghe Zhou and Yuyi Lu for assistance with the experimental setup in China. We also thank the Biodiversity Unit at the University of Oulu for providing space to carry out the work in Finland.

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
