## [Peer Review File · Proceedings of the Royal Society B: Biological Sciences]

Review History

RSPB-2020-2711.R0 (Original submission)

Review form: Reviewer 1

Recommendation

Accept with minor revision (please list in comments)

Scientific importance: Is the manuscript an original and important contribution to its field?

Good

General interest: Is the paper of sufficient general interest?

Excellent

Quality of the paper: Is the overall quality of the paper suitable?

Good

Is the length of the paper justified?

No

Should the paper be seen by a specialist statistical reviewer?

No

Do you have any concerns about statistical analyses in this paper? If so, please specify them explicitly in your report.

No

It is a condition of publication that authors make their supporting data, code and materials available - either as supplementary material or hosted in an external repository. Please rate, if applicable, the supporting data on the following criteria.

Is it accessible?

Yes

Is it clear?

No

Is it adequate?

No

Do you have any ethical concerns with this paper?

No

Comments to the Author

This is a straightforward paper that clearly demonstrates a highly relevant phenomenon: that training protocols taken as evidence for 'numerosity' in bees (and other animals) do not sufficiently control for alternative explanations based on assessment of continuous quantities. The experiment puts the number of items and the continuous quantities (edge length, convex hull, and spatial frequency) in the stimuli in direct competition and show the latter appear to be used by the bee for discrimination. A simple neural model based on spatial frequency detection is shown to reproduce bee behaviour that has been previously taken to demonstrate numerosity, including generalising 'more' or 'less' to novel stimuli pairs including zero. Overall the experiments are well designed, the analysis appropriate and the conclusions justified. Although this critique of numerosity experiments is not entirely novel, the demonstration here is particularly compelling. The outcome is important as it relates not only to bees but to similar numerosity tests used with other species.

Some specific minor comments for improvement:

Abstract: the final sentence is awkwardly phrased. Also, it might be more helpful to state here (i.e. in the abstract) more concretely what new ways of testing are being suggested (from the discussion these include: control tests for alternative cues, using cross-modal cues, analysing behavioural responses in more detail to detect underlying strategies, finding the neural substrate).

Introduction:

line 49 "Honeybees, along with many other animal species, have been shown to solve a variety of numeric-based tasks, from counting to basic math problems (e.g. [2-33])." The paper here and multiple times later makes 'bulk reference' to the set of papers [2-33]. This is frustrating as it is hard to know to what extent a specific sentence applies to all these papers. E.g. from the structure of the sentence above, I would assume papers 2-33 refer to honeybees, but in fact they refer to many animal species. Line 63 "By far, the most common method for testing numerical cognition in animals is to have subjects discriminate 2D visual displays with differing numbers of shapes (e.g. [2-33])." Do all these papers use this specific method? Line 68 "Although many studies have attempted to control for the use of some continuous properties, at least one or more continuous cues often still covary with numerosity, and are not tested for (e.g. [2-33]; figure 1)." It would be useful to have some kind of breakdown of which papers have controlled for

which continuous properties, and whether any have previously found that such properties are a confound, e.g. whether total area of the stimulus will be used by the animal if it is not controlled for. See also line 123 in the Results.

Results & Discussion

Line 129 on, the explanation of the neural net model could more clearly state (in the main text, not just the methods) what the 'three layers' consist of, i.e., seven 'sensory' neurons filtering for different spatial frequencies (subsequent to Fourier transform on the images), an output neuron providing a weighted summed response, and a reward signalling neuron that alters the weighting between the sensory neurons and the output during training with the experimental stimuli.

I also find this model not wholly compelling as a demonstration that the task can be solved by "a simple computational structure using only non-numerical information". A fourier transform is not necessarily a trivial image pre-processing step for a biological visual system. Moreover, one could propose instead a network with seven 'sensory' neurons each 'broadly tuned' to a different number of items in the stimulus (similarly extracted by some image pre-processing). This would likely also reproduce the results, this time using numerical information. The model is presented in the context of the question (line 129) "what explanation is simpler and more plausible: numerical or non-numerical processing?" The answer depends more on the assumptions about pre-processing, which are not discussed, than on the structure of the model.

Line 171 "we've found that practically no studies have tested for all continuous variables". Does 'practically no studies' mean 'no studies'? It is crucial to be clear here - if there is any study that *has* tested for all continuous variables, and still found numerosity, it needs to be highlighted and discussed.

The discussion, effectively from line 151 onward, makes a number of good points but could be better organised and more concise. For example, the idea of simultaneous testing and reference [62] occurs at line 188, but then the text goes on to discuss a different study, [34], and then returns to the same concept and reference [62] at line 225. The paragraph lines 237-242 seems wholly repetition, and the following paragraph largely unnecessary.

Review form: Reviewer 2

Recommendation

Major revision is needed (please make suggestions in comments)

Scientific importance: Is the manuscript an original and important contribution to its field?

Acceptable

General interest: Is the paper of sufficient general interest?

Good

Quality of the paper: Is the overall quality of the paper suitable?

Good

Is the length of the paper justified?

Yes

Should the paper be seen by a specialist statistical reviewer?

No

Do you have any concerns about statistical analyses in this paper? If so, please specify them explicitly in your report.

No

It is a condition of publication that authors make their supporting data, code and materials available - either as supplementary material or hosted in an external repository. Please rate, if applicable, the supporting data on the following criteria.

Is it accessible?

Yes

Is it clear?

Yes

Is it adequate?

Yes

Do you have any ethical concerns with this paper?

No

Comments to the Author

The paper by MaBoudi and colleagues presents evidence of the using of non-numerical cues (i.e., continuous variables, such as area, edge length and convex hull, that co-vary with numerosness) by bees trained to discriminate among different quantities. The authors trained two independent groups of bees to select the larger and the smaller quantity in the contrast, respectively. The stimuli were 2D elements, previously used in a previous study (Howard et al Science) investigating numerical abilities in honeybees. The authors suggested use of non-numerical cues to solve numerical discrimination in honeybees.

I think the paper is potentially interesting but there are a few major questions that need to be addressed before any decision on the suitability of this paper for publication can be reached.

Major comments

- The first issue is related to what seems to me a major inconsistency with the paper (and stimuli) they used for replication (Ref 22). The author reported that, to their knowledge, there is no studies on numerical abilities of animals that has considered the role of spatial frequency in quantity discrimination. However, I am puzzled because the stimuli used by the authors were the same as in a previous study by Howard and colleagues (Howard, S. R., Avarguès-Weber, A., Garcia, J. E., Greentree, A. D., & Dyer, A. G. (2018). Numerical ordering of zero in honey bees. *Science*, 360(6393), 1124-1126) and these authors claimed that the stimuli “were controlled for colour balance, spatial frequency, surface area, pattern, shape, and element sizes” (Fig. S2, Supplementary materials). Thus, there seems to be some inconsistency here. Moreover, if I read correctly the Howard et al paper no control of edge length and convex hull was there, only of spatial frequency. If the authors of the present ms. used the same stimuli used by Howard et al it would be not very surprising that when at test numerosity was the same bees used the remaining continuous cues. Even in the case in which continuous cues were in the opposite direction to the numerical difference, the results are not convincingly against the encoding of discrete numerosities during training, for given the amount of massive change in continuous physical variables (edge length, convex hull and spatial frequency) occurring at test it could well be that bees turned to the use of the latter.
- It would be important that the authors should provide the results of the two groups (bees trained on more-than and bees trained on less-than) separately, and that the effects at test of the different type of training are tested statistically in a separate way.
- It is not clear what kind of measurement the authors considered as dependent variable during the test phase, neither it is reported how many choices were scored during the test and the duration of this phase. Since previous studies on numerical abilities in honeybees used slightly

different test phases, with either the consideration of a fixed amount of time during which each interaction with the stimuli is considered or the scoring of a fixed number of choices, the authors should report accurately these details of the test phase.

Other Comments:

- The author might consider reporting the average number of training trials complete by the subjects to reach the 80% of accuracy, as well as mean \pm s.e.m of the test performances for a better understanding of the graph 2b.
- Line 58-61: it seems that some reference should be added to support the statement “[...]these results, along with other works suggesting honeybees and other animals are able to solve tasks in an unexpected ways[...]”.
- Line 296: the authors might consider change “contained” with “associated with”.
- Line 297: the authors might consider change “reminder trials” with “refresh trials”, since the latter is more commonly used in studies on numerical abilities.
- Figure 2c-f: the authors should consider to enlarge those figures since it is quite difficult to appreciate the stimuli represented on the x axis.
- The author might consider to propose a specific name for each of the test presented (as for instance, learning test, continuous generalization test, continuous incongruent test) in order to help the reader also in the interpretation of the graphs.
- The author wrote that all the continuous variables were tested simultaneously only once, in a study recently published (MaBouDi H, Dona HSG, Gatto E, Loukola OJ, Buckley E, Onoufriou PD, Skorupski P, Chittka L. 2020 Bumblebees use sequential scanning of countable items in visual patterns to solve numerosity tasks. *Integr. Comp. Biol.* (doi:10.1093/icb/icaa025)). I do not see how all continuous variables can be tested simultaneously, however. This is simply not possible. Looking at the paper I found that indeed the authors only presented stimuli with different element’s dimension, shape and colour at test. Thus, several other continuous variables were not controlled.
- Line 45: I think this should be substantiated by at least some general reference; in particular, as to the ‘innate’ part I believe the only direct evidence for that comes from studies in newborn chicks (see e.g. for a review: Vallortigara, G. (2017). *An animal’s sense of number. In “The nature and Development of Mathematics. Cross Disciplinary Perspective on Cognition, Learning and Culture”* (Adams, J.W., Barmby P., Mesoudi, A., eds.), pp. 43-65, Routledge, New York).
- Line 45: “Recent work” looks a bit weird, for the idea that numerosness encoding is based on magnitude dates back to classical work by Randy Gallistel and, after that, to the so-called ATOM Theory by Walsh. I think the authors should make an effort to provide a proper theoretical framework for the important issues they raised. In fact, even with insects there has been recent work arguing for generalization between the domains of discrete and continuous magnitudes that looks quite relevant to this paper (see Bortot et al (2020). *Transfer from number to size reveals abstract coding of magnitude in honeybees. iScience* 23, 101122 <https://doi.org/10.1016/j.isci.2020.101122>).
- Line 50: Consider quoting a recent general review on number sense in invertebrates (Bortot et al (2020). *A sense of number in invertebrates. Biochemical and Biophysical Research Communications*, <https://doi.org/10.1016/j.bbrc.2020.11.039>
- Line 70: Among the most convincing evidence with good control conditions consider however the following: Rugani et al 2010. *Imprinted numbers: Newborn chicks' sensitivity to number vs. continuous extent of objects they have been reared with. Developmental Science*, 13: 790-797; and in frogs: Stancher et al. (2015). *Numerical discrimination by frogs (Bombina orientalis). Animal Cognition*, 18: 219-229.
- Line 107: This seems to me not a logical conclusion: given the previous training, bees may be simply generalizing among different magnitudes as in the Bortot et al paper mentioned above.
- Para. 110-119: Here I think disentangling data for learning “more” or “less” would be crucial. Also, I am wondering whether a different explanation could be as follows. Let’s suppose that during training bees encode different dimensions of magnitude, both discrete (number) and continuous (edge length, convex hull, and spatial frequency). Given that at test the authors

introduced massive changes in continuous variables (at least 3 dimensions vs. the only 1 of discrete) it may appear not surprising that bees tended to use continuous variables.

- Line 135: As I stated above this statement contrasts strikingly with what is reported in the Supplementary materials of the Science paper, in which spatial frequency seems indeed to have been controlled for.

Decision letter (RSPB-2020-2711.R0)

09-Dec-2020

Dear Dr Solvi:

Your manuscript has now been peer reviewed and the reviews have been assessed by an Associate Editor. The reviewers' comments (not including confidential comments to the Editor) and the comments from the Associate Editor are included at the end of this email for your reference. As you will see, the reviewers and the associate editor are generally positive about your manuscript, but have raised some concerns that we would like to invite you to address in a revision. These concerns are detailed below, but in short, both highlight some concerns about your treatment of the existing literature, and in particular, reviewer 2 highlights two papers that require a much more careful assessment relative to the current paper (esp Howard et al, 2018). This reviewer also suggests that one possible explanation for your results is that, given the massive number of aspects of the stimuli that change, the bees simply chose one of these continuous quantities to focus on when making the discrimination. Both reviewers also note that your manuscript is very difficult to follow, and request clarifications in the introduction and methods in numerous places. For instance, as reviewer 1 mentions, it would be helpful to explicitly detail what exactly you are testing so that the reader can more easily follow it. Finally, both reviewers request additional statistical comparisons, and additional clarification around your code and supplemental data (which is not currently loading properly). Each reviewers' comments, as well as those of the AE, are available in full below, minus any confidential comments to the editor.

Research ethics:

Use of animals and field studies:

It is a condition of publication that you make available the data and research materials supporting the results in the article. Please see our Data Sharing Policies (<https://royalsociety.org/journals/authors/author-guidelines/#data>). Datasets should be deposited in an appropriate publicly available repository and details of the associated accession number, link or DOI to the datasets must be included in the Data Accessibility section of the article (<https://royalsociety.org/journals/ethics-policies/data-sharing-mining/>). Reference(s) to datasets should also be included in the reference list of the article with DOIs (where available).

Please submit a copy of your revised paper within three weeks. If we do not hear from you within this time your manuscript will be rejected. If you are unable to meet this deadline please let us know as soon as possible, as we may be able to grant a short extension.

Best wishes,
 Dr Sarah Brosnan
 Editor, Proceedings B
 mailto: proceedingsb@royalsociety.org

Associate Editor
 Board Member: 1
 Comments to Author:

Two expert reviewers have provided comments on your manuscript. Both have generally positive views of the study, and agree it has general scientific important and interest. However, both highlight several areas that should be addressed. These mostly pertain to the clarity of the text and referencing, which is in places difficult to interpret. Reviewer one highlights some particular examples where the phrasing, methodological descriptions or referencing can be improved. Reviewer two also highlights some examples that relate to the methodology. They also request some additional statistical comparisons. I agree with the reviewers, I found the paper interesting but not particularly reader-friendly. It might make the paper more generally accessible to unpack the key messages and past literature in more detail in the introduction, and to add some second level subheadings to the R&D and Methods. One reviewer also commented that it was not easy to follow how to use the code provided to reconstruct and re-run the model. Some clearer structure and guidance, or a 'read me' file might be useful. The SourceData.xls file also doesn't currently load.

Reviewer(s)' Comments to Author:

Referee: 1
 Comments to the Author(s)

This is a straightforward paper that clearly demonstrates a highly relevant phenomenon: that training protocols taken as evidence for 'numerosity' in bees (and other animals) do not sufficiently control for alternative explanations based on assessment of continuous quantities. The experiment puts the number of items and the continuous quantities (edge length, convex hull, and spatial frequency) in the stimuli in direct competition and show the latter appear to be used by the bee for discrimination. A simple neural model based on spatial frequency detection is shown to reproduce bee behaviour that has been previously taken to demonstrate numerosity, including generalising 'more' or 'less' to novel stimuli pairs including zero. Overall the experiments are well designed, the analysis appropriate and the conclusions justified. Although this critique of numerosity experiments is not entirely novel, the demonstration here is particularly compelling. The outcome is important as it relates not only to bees but to similar numerosity tests used with other species.

Some specific minor comments for improvement:

Abstract: the final sentence is awkwardly phrased. Also, it might be more helpful to state here (i.e. in the abstract) more concretely what new ways of testing are being suggested (from the discussion these include: control tests for alternative cues, using cross-modal cues, analysing behavioural responses in more detail to detect underlying strategies, finding the neural substrate).

Introduction:

line 49 "Honeybees, along with many other animal species, have been shown to solve a variety of numeric-based tasks, from counting to basic math problems (e.g. [2-33])."

The paper here and multiple times later makes ‘bulk reference’ to the set of papers [2-33]. This is frustrating as it is hard to know to what extent a specific sentence applies to all these papers. E.g. from the structure of the sentence above, I would assume papers 2-33 refer to honeybees, but in fact they refer to many animal species. Line 63 “By far, the most common method for testing numerical cognition in animals is to have subjects discriminate 2D visual displays with differing numbers of shapes (e.g.[2-33]).” Do all these papers use this specific method? Line 68 “Although many studies have attempted to control for the use of some continuous properties, at least one or more continuous cues often still covary with numerosity, and are not tested for (e.g. [2-33]; figure 1).” It would be useful to have some kind of breakdown of which papers have controlled for which continuous properties, and whether any have previously found that such properties are a confound, e.g. whether total area of the stimulus will be used by the animal if it is not controlled for. See also line 123 in the Results.

Results & Discussion

Line 129 on, the explanation of the neural net model could more clearly state (in the main text, not just the methods) what the ‘three layers’ consist of, i.e., seven ‘sensory’ neurons filtering for different spatial frequencies (subsequent to Fourier transform on the images), an output neuron providing a weighted summed response, and a reward signalling neuron that alters the weighting between the sensory neurons and the output during training with the experimental stimuli.

I also find this model not wholly compelling as a demonstration that the task can be solved by “a simple computational structure using only non-numerical information”. A fourier transform is not necessarily a trivial image pre-processing step for a biological visual system. Moreover, one could propose instead a network with seven ‘sensory’ neurons each ‘broadly tuned’ to a different number of items in the stimulus (similarly extracted by some image pre-processing). This would likely also reproduce the results, this time using numerical information. The model is presented in the context of the question (line 129) “what explanation is simpler and more plausible: numerical or non-numerical processing?” The answer depends more on the assumptions about pre-processing, which are not discussed, than on the structure of the model.

Line 171 “we’ve found that practically no studies have tested for all continuous variables”. Does ‘practically no studies’ mean ‘no studies’? It is crucial to be clear here - if there is any study that *has* tested for all continuous variables, and still found numerosity, it needs to be highlighted and discussed.

The discussion, effectively from line 151 onward, makes a number of good points but could be better organised and more concise. For example, the idea of simultaneous testing and reference [62] occurs at line 188, but then the text goes on to discuss a different study, [34], and then returns to the same concept and reference [62] at line 225. The paragraph lines 237-242 seems wholly repetition, and the following paragraph largely unnecessary.

Referee: 2

Comments to the Author(s)

The paper by MaBoudi and colleagues presents evidence of the using of non-numerical cues (i.e., continuous variables, such as area, edge length and convex hull, that co-vary with numerosity) by bees trained to discriminate among different quantities. The authors trained two independent groups of bees to select the larger and the smaller quantity in the contrast, respectively. The stimuli were 2D elements, previously used in a previous study (Howard et al Science) investigating numerical abilities in honeybees. The authors suggested use of non-numerical cues to solve numerical discrimination in honeybees.

I think the paper is potentially interesting but there are a few major questions that need to be addressed before any decision on the suitability of this paper for publication can be reached.

Major comments

- The first issue is related to what seems to me a major inconsistency with the paper (and stimuli) they used for replication (Ref 22). The author reported that, to their knowledge, there is no studies on numerical abilities of animals that has considered the role of spatial frequency in quantity discrimination. However, I am puzzled because the stimuli used by the authors were the same as in a previous study by Howard and colleagues (Howard, S. R., Avarguès-Weber, A., Garcia, J. E., Greentree, A. D., & Dyer, A. G. (2018). Numerical ordering of zero in honey bees. *Science*, 360(6393), 1124-1126) and these authors claimed that the stimuli “were controlled for colour balance, spatial frequency, surface area, pattern, shape, and element sizes” (Fig. S2, Supplementary materials). Thus, there seems to be some inconsistency here. Moreover, if I read correctly the Howard et al paper no control of edge length and convex hull was there, only of spatial frequency. If the authors of the present ms. used the same stimuli used by Howard et al it would be not very surprising that when at test numerosity was the same bees used the remaining continuous cues. Even in the case in which continuous cues were in the opposite direction to the numerical difference, the results are not convincingly against the encoding of discrete numerosities during training, for given the amount of massive change in continuous physical variables (edge length, convex hull and spatial frequency) occurring at test it could well be that bees turned to the use of the latter.

- It would be important that the authors should provide the results of the two groups (bees trained on more-than and bees trained on less-than) separately, and that the effects at test of the different type of training are tested statistically in a separate way.

- It is not clear what kind of measurement the authors considered as dependent variable during the test phase, neither it is reported how many choices were scored during the test and the duration of this phase. Since previous studies on numerical abilities in honeybees used slightly different test phases, with either the consideration of a fixed amount of time during which each interaction with the stimuli is considered or the scoring of a fixed number of choices, the authors should report accurately these details of the test phase.

Other Comments:

- The author might consider reporting the average number of training trials complete by the subjects to reach the 80% of accuracy, as well as mean \pm s.e.m of the test performances for a better understanding of the graph 2b.

- Line 58-61: it seems that some reference should be added to support the statement “[...]these results, along with other works suggesting honeybees and other animals are able to solve tasks in an unexpected ways[...]”.

- Line 296: the authors might consider change “contained” with “associated with”.

- Line 297: the authors might consider change “reminder trials” with “refresh trials”, since the latter is more commonly used in studies on numerical abilities.

- Figure 2c-f: the authors should consider to enlarge those figures since it is quite difficult to appreciate the stimuli represented on the x axis.

- The author might consider to propose a specific name for each of the test presented (as for instance, learning test, continuous generalization test, continuous incongruent test) in order to help the reader also in the interpretation of the graphs.

- The author wrote that all the continuous variables were tested simultaneously only once, in a study recently published (MaBouDi H, Dona HSG, Gatto E, Loukola OJ, Buckley E, Onoufriou PD, Skorupski P, Chittka L. 2020 Bumblebees use sequential scanning of countable items in visual patterns to solve numerosity tasks. *Integr. Comp. Biol.* (doi:10.1093/icb/icaa025)). I do not see how all continuous variables can be tested simultaneously, however. This is simply not possible. Looking at the paper I found that indeed the authors only presented stimuli with different element’s dimension, shape and colour at test. Thus, several other continuous variables were not controlled.

- Line 45: I think this should be substantiated by at least some general reference; in particular, as to the ‘innate’ part I believe the only direct evidence for that comes from studies in newborn chicks (see e.g. for a review: Vallortigara, G. (2017). An animal’s sense of number. In “The nature and Development of Mathematics. Cross Disciplinary Perspective on Cognition, Learning and Culture” (Adams, J.W., Barmby P., Mesoudi, A., eds.), pp. 43-65, Routledge, New York.

- Line 45: “Recent work” looks a bit weird, for the idea that numerosness encoding is based on magnitude dates back to classical work by Randy Gallistel and, after that, to the so-called ATOM Theory by Walsh. I think the authors should make an effort to provide a proper theoretical framework for the important issues they raised. In fact, even with insects there has been recent work arguing for generalization between the domains of discrete and continuous magnitudes that looks quite relevant to this paper (see Bortot et al (2020). Transfer from number to size reveals abstract coding of magnitude in honeybees. *iScience* 23, 101122 <https://doi.org/10.1016/j.isci.2020.101122>).
- Line 50: Consider quoting a recent general review on number sense in invertebrates (Bortot et al (2020). A sense of number in invertebrates. *Biochemical and Biophysical Research Communications*, <https://doi.org/10.1016/j.bbrc.2020.11.039>
- Line 70: Among the most convincing evidence with good control conditions consider however the following: Rugani et al 2010. Imprinted numbers: Newborn chicks' sensitivity to number vs. continuous extent of objects they have been reared with. *Developmental Science*, 13: 790-797; and in frogs: Stancher et al. (2015). Numerical discrimination by frogs (*Bombina orientalis*). *Animal Cognition*, 18: 219-229.
- Line 107: This seems to me not a logical conclusion: given the previous training, bees may be simply generalizing among different magnitudes as in the Bortot et al paper mentioned above.
- Para. 110-119: Here I think disentangling data for learning “more” or “less” would be crucial. Also, I am wondering whether a different explanation could be as follows. Let’s suppose that during training bees encode different dimensions of magnitude, both discrete (number) and continuous (edge length, convex hull, and spatial frequency). Given that at test the authors introduced massive changes in continuous variables (at least 3 dimensions vs. the only 1 of discrete) it may appear not surprising that bees tended to use continuous variables.
- Line 135: As I stated above this statement contrasts strikingly with what is reported in the Supplementary materials of the Science paper, in which spatial frequency seems indeed to have been controlled for.

Author's Response to Decision Letter for (RSPB-2020-2711.R0)

See Appendix A.

RSPB-2020-2711.R1 (Revision)

Review form: Reviewer 1

Recommendation

Accept with minor revision (please list in comments)

Scientific importance: Is the manuscript an original and important contribution to its field?

Excellent

General interest: Is the paper of sufficient general interest?

Excellent

Quality of the paper: Is the overall quality of the paper suitable?

Excellent

Is the length of the paper justified?

Yes

Should the paper be seen by a specialist statistical reviewer?

No

Do you have any concerns about statistical analyses in this paper? If so, please specify them explicitly in your report.

No

It is a condition of publication that authors make their supporting data, code and materials available - either as supplementary material or hosted in an external repository. Please rate, if applicable, the supporting data on the following criteria.

Is it accessible?

Yes

Is it clear?

Yes

Is it adequate?

Yes

Do you have any ethical concerns with this paper?

No

Comments to the Author

Overall I am satisfied with the changes made in response to my original review. I have the following minor changes to suggest.

Abstract - The first sentence of the abstract is rather awkwardly phrased, and seems overly general and not really necessary. The term 'numeric-based' in the second sentence is unclear, and the sense could be more clearly conveyed, e.g. by "We examined how bees solve a decision task that uses stimuli commonly found in numerical cognition studies". I also think the phrase "a simple network model containing just nine elements" is not an adequate reflection of the model; it would be better to say something like "a model using biologically plausible spatial frequency filtering and a simple associative rule". 'Nine elements' is not meaningful when the complexity of each element is unknown.

I appreciate that the authors have now clarified why pre-processing using frequency filtering is more plausible than 'numeric' filtering, but there is still some difference between (local) Gabor-like filters and (global) Fourier analysis that mean the model is not as 'simple' as they repeatedly claim. E.g how many simple and complex cells might be needed, and their output integrated in what way (in subsequent layers?), to produce the same response as one 'element' in their model which is tuned to a preferred Fourier frequency for the whole image? It also seems unnecessary to describe the decision element as "a neuron in the mushroom bodies" given the abstraction level of this model.

Review form: Reviewer 2

Recommendation

Accept with minor revision (please list in comments)

Scientific importance: Is the manuscript an original and important contribution to its field?

Good

General interest: Is the paper of sufficient general interest?

Good

Quality of the paper: Is the overall quality of the paper suitable?

Good

Is the length of the paper justified?

Yes

Should the paper be seen by a specialist statistical reviewer?

No

Do you have any concerns about statistical analyses in this paper? If so, please specify them explicitly in your report.

No

It is a condition of publication that authors make their supporting data, code and materials available - either as supplementary material or hosted in an external repository. Please rate, if applicable, the supporting data on the following criteria.

Is it accessible?

Yes

Is it clear?

Yes

Is it adequate?

Yes

Do you have any ethical concerns with this paper?

No

Comments to the Author

I believe the authors have addressed adequately all my concerns, and that the paper deserves to be published. I have only one final issue.

On p. 3 lines 81-82 the Authors stated that they found no studies that tested for all continuous variables. It seems to me, however, that the paper by Bortot et al (2020) Transfer from number to size reveals abstract coding of magnitude in honeybees. *iScience* 23, 101122 <https://doi.org/10.1016/j.isci.2020.101122>, that they did not cite, is in fact providing an example, of such a control for the authors of this paper checked for overall area, perimeter (contour length), convex hull and density. They also balanced the presence of the largest element (a third of the time it was in the smaller number group, a third of the time in the larger number group and a third of the time was present in both).

The only parameter they did not check for during training was spatial frequency, however, if the bees were using this parameter one should have expected to observe that in the space (size) generalization test: i.e., that the bees trained to choose the largest number, which could contain overall the relatively smallest dots, would have had to choose the smaller elements at test, and vice versa those trained to choose the smaller number that could contain overall the relatively largest elements in half of the cases, they should have chosen the largest elements at test. The opposite was observed. Thus, as far as I can judge, this paper does in fact provide a control for all continuous variables.

Decision letter (RSPB-2020-2711.R1)

12-Jan-2021

Dear Dr Solvi

I am pleased to inform you that your manuscript RSPB-2020-2711.R1 entitled "Non-numerical strategies used by bees to solve numerical cognition tasks" has been accepted for publication in Proceedings B pending minor revision suggested by the reviewers. Therefore, I invite you to respond to the referee(s)' comments and revise your manuscript. Because the schedule for publication is very tight, it is a condition of publication that you submit the revised version of your manuscript within 7 days. If you do not think you will be able to meet this date please let us know.

- 4) A media summary: a short non-technical summary (up to 100 words) of the key findings/importance of your manuscript.
- 5) Data accessibility section and data citation

[http://datadryad.org/submit?journalID=RSPB&manu=\(Document not available\)](http://datadryad.org/submit?journalID=RSPB&manu=(Document+not+available)) which will take you to your unique entry in the Dryad repository. If you have already submitted your data to dryad you can make any necessary revisions to your dataset by following the above link. Please see <https://royalsociety.org/journals/ethics-policies/data-sharing-mining/> for more details.

Sincerely,
Dr Sarah Brosnan
Editor, Proceedings B
<mailto:proceedingsb@royalsociety.org>

Associate Editor:

Board Member: 1

Comments to Author:

Both original reviewers have (within 24 hours of our request!) re-assessed your manuscript and are positive about both your study and the revisions you have made to it. They have suggested some minor comments that would be good to address, but which will not require further review.

Referee: 1

Comments to the Author(s)

Overall I am satisfied with the changes made in response to my original review. I have the following minor changes to suggest.

Abstract - The first sentence of the abstract is rather awkwardly phrased, and seems overly general and not really necessary. The term 'numeric-based' in the second sentence is unclear, and the sense could be more clearly conveyed, e.g. by "We examined how bees solve a decision task that uses stimuli commonly found in numerical cognition studies". I also think the phrase "a simple network model containing just nine elements" is not an adequate reflection of the model; it would be better to say something like "a model using biologically plausible spatial frequency

filtering and a simple associative rule". 'Nine elements' is not meaningful when the complexity of each element is unknown.

I appreciate that the authors have now clarified why pre-processing using frequency filtering is more plausible than 'numeric' filtering, but there is still some difference between (local) Gabor-like filters and (global) Fourier analysis that mean the model is not as 'simple' as they repeatedly claim. E.g how many simple and complex cells might be needed, and their output integrated in what way (in subsequent layers?), to produce the same response as one 'element' in their model which is tuned to a preferred Fourier frequency for the whole image? It also seems unnecessary to describe the decision element as "a neuron in the mushroom bodies" given the abstraction level of this model.

Referee: 2

Comments to the Author(s)

I believe the authors have addressed adequately all my concerns, and that the paper deserves to be published. I have only one final issue.

On p. 3 lines 81-82 the Authors stated that they found no studies that tested for all continuous variables. It seems to me, however, that the paper by Bortot et al (2020) Transfer from number to size reveals abstract coding of magnitude in honeybees. *iScience* 23, 101122

<https://doi.org/10.1016/j.isci.2020.101122>, that they did not cite, is in fact providing an example, of such a control for the authors of this paper checked for overall area, perimeter (contour length), convex hull and density. They also balanced the presence of the largest element (a third of the time it was in the smaller number group, a third of the time in the larger number group and a third of the time was present in both).

The only parameter they did not check for during training was spatial frequency, however, if the bees were using this parameter one should have expected to observe that in the space (size) generalization test: i.e., that the bees trained to choose the largest number, which could contain overall the relatively smallest dots, would have had to choose the smaller elements at test, and vice versa those trained to choose the smaller number that could contain overall the relatively largest elements in half of the cases, they should have chosen the largest elements at test. The opposite was observed. Thus, as far as I can judge, this paper does in fact provide a control for all continuous variables.

Author's Response to Decision Letter for (RSPB-2020-2711.R1)

See Appendix B.

Decision letter (RSPB-2020-2711.R2)

18-Jan-2021

Dear Dr Solvi

I am pleased to inform you that your manuscript entitled "Non-numerical strategies used by bees to solve numerical cognition tasks" has been accepted for publication in *Proceedings B*.

Open Access

Paper charges

Sincerely,

Proceedings B

Appendix A

12/25/2020

Dear Dr Brosnan,

Please find attached a thoroughly revised version of the manuscript **Non-numerical strategies used by bees to solve numerical cognition tasks** (RSPB-2020-2711), which we would like to resubmit as a *Research Article to Proc B*.

On December 9, 2020, you had sent us an email stating your interest in a resubmission of our manuscript once we were able to fully address the concerns raised by the reviewers. The reviewers' comments have been very helpful in improving the manuscript. All suggestions have been fully addressed. In particular:

- (1) To make the manuscript more concise and easier to follow, based on both Referees' comments, we have made significant changes throughout and provided clarifications in the Introduction and Materials and Methods sections.
- (2) We provide complete details on the glmms now performed that address concerns raised by both referees. In particular, as requested by Referee #2, glmm results show that the rule learned by bees (more-than or less-than) had no effect on their test performance. We also report the performance mean and s.e.m of both groups as requested by Referee #2.
- (3) Second level subheadings have been added throughout to make the manuscript easier to follow.
- (4) In response to Referee #2, we explain that Howard et al. 2018 did not perform any analyses to test whether spatial frequency covaried with numerosity and they provide no real comparison or explanation of the spatial frequency data they present in their Supplemental Materials. In contrast, we provide details on how we calculated spatial frequency and provide correlation analyses showing that spatial frequency (as well as convex hull and edge length) covary with number in their stimuli.
- (5) In response to Referee #2's suggestion that bees may have learned number along with continuous cues, we clarify that there is not sufficient evidence for this explanation provided by the methods commonly employed in numerical cognition studies. This is the very point of our paper and is supported by our results. As pointed out by Referee #1, our results clearly show that bees learned continuous cues and do not require numerosity to solve the task.
- (6) We have added clarification on our model in the main text as requested by Referee #1, added a readme file to help readers find and re-run the code for the model, and verified that the source data file will download from Figshare (note: the preview does not load).

Below, we first repeat the referee comments in bold and then follow each with our answers in italics. Thank you for your time and effort and we hope that you will find the new version acceptable for publication in *Proc B*.

Kind regards,

Cwyn Solvi

Referee #1

This is a straightforward paper that clearly demonstrates a highly relevant phenomenon: that training protocols taken as evidence for ‘numerosity’ in bees (and other animals) do not sufficiently control for alternative explanations based on assessment of continuous quantities. The experiment puts the number of items and the continuous quantities (edge length, convex hull, and spatial frequency) in the stimuli in direct competition and show the latter appear to be used by the bee for discrimination. A simple neural model based on spatial frequency detection is shown to reproduce bee behaviour that has been previously taken to demonstrate numerosity, including generalising ‘more’ or ‘less’ to novel stimuli pairs including zero. Overall the experiments are well designed, the analysis appropriate and the conclusions justified. Although this critique of numerosity experiments is not entirely novel, the demonstration here is particularly compelling. The outcome is important as it relates not only to bees but to similar numerosity tests used with other species.

Thank you!

Some specific minor comments for improvement:

Abstract: the final sentence is awkwardly phrased. Also, it might be more helpful to state here (i.e. in the abstract) more concretely what new ways of testing are being suggested (from the discussion these include: control tests for alternative cues, using cross-modal cues, analysing behavioural responses in more detail to detect underlying strategies, finding the neural substrate).

We have deleted the previous final sentence of the abstract and replaced it with the following: “We suggest ways of better assessing numerical cognition in non-speaking animals, including assessing the use of all alternative cues in one test, using cross-modal cues, analysing behavioural responses to detect underlying strategies, and finding the neural substrate.”

Introduction:

line 49 “Honeybees, along with many other animal species, have been shown to solve a variety of numeric-based tasks, from counting to basic math problems (e.g. [2–33]).” The paper here and multiple times later makes ‘bulk reference’ to the set of papers [2-33]. This is frustrating as it is hard to know to what extent a specific sentence applies to all these papers. E.g. from the structure of the sentence above, I would assume papers 2-33 refer to honeybees, but in fact they refer to many animal species. Line 63 “By far, the most common method for testing numerical cognition in animals is to have subjects discriminate 2D visual displays with differing numbers of shapes (e.g.[2–33]).” Do all these papers use this specific method? Line 68 “Although many studies have attempted to control for the use of some continuous properties, at least one or more continuous cues often still covary with numerosity, and are not tested for (e.g. [2–33]; figure 1).” It would be useful to have some kind of breakdown of which papers have controlled

for which continuous properties, and whether any have previously found that such properties are a confound, e.g. whether total area of the stimulus will be used by the animal if it is not controlled for. See also line 123 in the Results.

We apologise for the lack of clarity in our previous version. We have now improved clarity in the following ways:

We changed the sentence on previous line 49 (now on line 56) to “Numerical cognition has been claimed in a large number of animal species (e.g. [8–39]), suggesting that a sense of number is widespread (for reviews see [40–42]).”

Previous line 63 (now on line 57) has now been changed to “By far, the most common method for testing numerical cognition in non-verbal animals is to have subjects discriminate 2D visual displays with differing numbers of shapes (Fig 1; [8–39] all used this design).”

Previous line 123 in Results has been deleted.

We have also added the following sentence with references to line 74 to provide a couple examples of papers that have shown continuous variables being used by animals: “Further, several works show that animals use non-numerical cues to solve numeric-based tasks when not controlled for, e.g. size of elements [53], total area [54], and convex hull [55], and even when they are controlled (e.g. [56]; see Discussion).” We believe that these few examples provide the reader with enough information and references for the reader. We believe providing an extensive breakdown of what each of the very many numerical cognition papers control would be outside the scope of the current manuscript and would be more appropriate for a review. We hope you agree and find these added lines and references sufficient.

Results & Discussion

Line 129 on, the explanation of the neural net model could more clearly state (in the main text, not just the methods) what the ‘three layers’ consist of, i.e., seven ‘sensory’ neurons filtering for different spatial frequencies (subsequent to Fourier transform on the images), an output neuron providing a weighted summed response, and a reward signalling neuron that alters the weighting between the sensory neurons and the output during training with the experimental stimuli.

In the main text, on line 283, we now state “Seven elements acted as sensory neurons that encoded spatial frequency in the visual lobe and which projected frequency information to the eighth element, a single decision neuron in the mushroom bodies (high-level sensory integration centres involved in learning and memory). Synaptic weights between the sensory neurons and decision neuron were adjusted according to the activation (by presentation of stimuli) of the ninth element, a reinforcement neuron, based on the specific learning rule (more-than or less-than).”

I also find this model not wholly compelling as a demonstration that the task can be solved by “a simple computational structure using only non-numerical information”. A Fourier transform is not necessarily a trivial image pre-processing step for a biological visual system. Moreover, one could propose instead a network with seven ‘sensory’ neurons each ‘broadly tuned’ to a different number of items in the stimulus (similarly extracted by some image pre-processing). This would likely also reproduce the results, this time using numerical information. The model is presented in the context of the question (line 129) “what explanation is simpler and more plausible: numerical or non-numerical processing?” The answer depends more on the assumptions about pre-processing, which are not discussed, than on the structure of the model.

It is not necessary that the brain use Fourier transformation to extract frequency information from the visual input. For instance, the Gabor-like receptive field of simple and complex cells in the early visual system of primates filter and encode visual features such as orientation and spatial frequency. The same spatial frequency encoding schema is proposed in the insect brain. We clarify this now on line 164 where we state “Our model utilizes spatial frequency encoding that is supported by bees’ ability to discriminate visual patterns based on spatial frequency [49,50] and observed neurons in the visual lobe of insects that provide a mechanism of frequency coding [61,62]. Analogous to the spatial frequency coding in primates [63,64], bees may use Gabor-like filters in their visual lobe to extract spatial frequency information from visual stimuli [65].”

Further, numerical estimation is a type of concept learning that requires a rule to be applied across stimuli, independent of physical features of those stimuli. Concept learning of any type is understood to require more computational complexity than discrimination of simple physical features [1]. It is proposed that to learn and process numerical information a separate multi-layered learning process must be at work on the top of the sensory neurons [2,3]. This must be done from the population activity of sensory neurons that are already varying from stimuli to stimuli even with the same number of elements. Thus, a model capable of learning numerosity will by default require more layers of processing and will be more complex than a proposed model that utilizes only the magnitude of continuous features.

1. Zentall TR, Wasserman EA, Lazareva OF, Thompson RKR, Rattermann MJ. 2008 Concept learning in animals. *Comp. Cogn. Behav. Rev.* **3**, 13–45. (doi:10.3819/ccbr.2008.30002)
2. Zorzi M, Testolin A. 2018 An emergentist perspective on the origin of number sense. *Philos. Trans. R. Soc. B Biol. Sci.* **373**, 20170043. (doi:10.1098/rstb.2017.0043)
3. Testolin A. 2020 The Challenge of Modeling the Acquisition of Mathematical Concepts. *Front. Hum. Neurosci.* **14**. (doi:10.3389/fnhum.2020.00100)

Line 171 “we’ve found that practically no studies have tested for all continuous

variables”. Does ‘practically no studies’ mean ‘no studies’? It is crucial to be clear here - if there is any study that *has* tested for all continuous variables, and still found numerosity, it needs to be highlighted and discussed.

We apologise for the confusing phrasing. We now say “no studies”. In the Discussion section we now highlight our own previous work where we tested for all continuous cues in one unrewarded test. Importantly, although our results suggest that bees did not use continuous cues, we explain that other non-numerical strategies could still be at play. On line 339 we state “It will also not suffice to test for continuous cues separately because animals may learn multiple redundant cues and use those available when others are not [73–78]. Testing all continuous variables and numerosity simultaneously, i.e. within one test, can help determine if continuous variables have been learned. In one of our recent works, examining how bumblebees solved a numeric-based task, we assessed the use of continuous cues within one unrewarded test [79]. Here, bees were shown 10 stimuli during one unrewarded test with different numbers of elements and levels of continuous cues. We chose the characteristics of different stimuli so that the bees’ choices for some over others would reveal whether or not they had learned and used specific continuous cues to solve the task. For example, two displays both contained the same number of elements, but the elements in one of the displays had a greater edge-length. Bees chose these two displays equally in the test, suggesting they did not use edge length. However, if they had performed well on the test (i.e. more often chose stimuli based on the numerosity rule they had been trained) but had chosen one of these two stimuli significantly more than the other, this would suggest bees had learned and used edge-length instead of numerosity. We provided pairs of stimuli that varied in this way for edge-length, area, convex hull, spatial frequency and illusionary contour (Area was kept constant throughout training and tests and therefore did not need to be tested). We must keep in mind, as pointed out above, that even when this type of design suggests continuous cues were not used, as it had in our work, other strategies could still be used. Although bees’ behaviour [79] indicated some form of counting, the bumblebees could have used working spatial memory to avoid recently visited shapes (cf. “inhibition of return” [80,81]). Therefore, it is possible that bees discriminated stimuli based on duration of time taken to scan all shapes within a display, or perhaps by an accumulator mechanism responding to visual changes as they scanned past each shape [69]. Either of these possible strategies do not require a true sense of number.”

The discussion, effectively from line 151 onward, makes a number of good points but could be better organised and more concise. For example, the idea of simultaneous testing and reference [62] occurs at line 188, but then the text goes on to discuss a different study, [34], and then returns to the same concept and reference [62] at line 225. The paragraph lines 237-242 seems wholly repetition, and the following paragraph largely unnecessary.

We have changed the Discussion section to be more concise. We now have a summary/interpretation paragraph followed by a short section discussing why commonly used methods will not work to control for continuous cues, and end with a section

discussing the methods we feel are best to assess numerical cognition in non-verbal animals.

Thank you also for pointing out the incorrect citation. It should have been the same reference and has not been corrected.

We have also deleted the final two paragraphs per your suggestion.

Referee #2

The paper by MaBoudi and colleagues presents evidence of the using of non-numerical cues (i.e., continuous variables, such as area, edge length and convex hull, that co-vary with numerosness) by bees trained to discriminate among different quantities. The authors trained two independent groups of bees to select the larger and the smaller quantity in the contrast, respectively. The stimuli were 2D elements, previously used in a previous study (Howard et al Science) investigating numerical abilities in honeybees. The authors suggested use of non-numerical cues to solve numerical discrimination in honeybees. I think the paper is potentially interesting but there are a few major questions that need to be addressed before any decision on the suitability of this paper for publication can be reached.

Major comments

- **The first issue is related to what seems to me a major inconsistency with the paper (and stimuli) they used for replication (Ref 22). The author reported that, to their knowledge, there is no studies on numerical abilities of animals that has considered the role of spatial frequency in quantity discrimination. However, I am puzzled because the stimuli used by the authors were the same as in a previous study by Howard and colleagues (Howard, S. R., Avarguès-Weber, A., Garcia, J. E., Greentree, A. D., & Dyer, A. G. (2018). Numerical ordering of zero in honey bees. *Science*, 360(6393), 1124-1126) and these authors claimed that the stimuli “were controlled for colour balance, spatial frequency, surface area, pattern, shape, and element sizes” (Fig. S2, Supplementary materials). Thus, there seems to be some inconsistency here. Moreover, if I read correctly the Howard et al paper no control of edge length and convex hull was there, only of spatial frequency.**

Howard et al. 2018 claim in their main text that “The spatial frequencies of stimuli are also ruled out as a potential explanation for results”. To support this, in Supplemental Materials they provide “a spatial frequency plot, a power spectrum plot, and an intensity plot” for all 97 stimuli used. However, no measurements were reported or comparisons made outside of simply stating “The power spectra of the non-zero stimuli (numbered) are different from the spectrum of the empty set stimulus”. Please also note that their power spectra plots are illegible and not explained.

In contrast, we now report on line 154 how we calculated the spatial frequency of the stimuli: “To calculate the spatial frequency of the training and test stimuli, a two-dimensional Fourier transform on each image was performed, followed by a power

spectrum calculation as the square amplitude of the Fourier transform and averaged over orientation [60]. The actual power over all frequencies was then measured by calculating the area under the curve of the radially averaged power spectrum.”

Also, on line 225 we provide the results of correlation analyses of the power spectrum plots’ data produced from each of the stimuli used in the original experiment: “But, similar to many other numerical cognition studies, edge-length (Spearman correlation: $\rho=0.93$, $p=1.00e-40$), convex hull (Spearman correlation: $\rho=0.44$, $p=4.88e-6$), and spatial frequency (Spearman correlation: $\rho=0.92$, $p=1.00e-40$) covaried with number (figure 1f-j).” These results are visualized and explained in Figure 1f-j.

If the authors of the present ms. used the same stimuli used by Howard et al it would be not very surprising that when at test numerosity was the same bees used the remaining continuous cues. Even in the case in which continuous cues were in the opposite direction to the numerical difference, the results are not convincingly against the encoding of discrete numerosities during training, for given the amount of massive change in continuous physical variables (edge length, convex hull and spatial frequency) occurring at test it could well be that bees turned to the use of the latter.

We apologise for the lack of clarity in our previous version of the manuscript. Our intended message is that by using these common methods, we cannot determine whether bees learned numerosity. Your proposed explanation only holds true if bees learned continuous cues during training. Indeed, our results show that they did learn continuous cues. Bees may have learned numerosity, but we have no way of knowing this using this type of design. We hope that our thoroughly revised manuscript is much clearer on this. For example, on line 311, in the Discussion we now state “We are not suggesting that all numerical cognition studies are wrong or that no animal has numerical cognition. We show, however, that in a task using a 2D visual display set with differing number of shapes, non-numerical cues can be learned, they dominate over numerosity when equal to or set in opposition to number of elements, and they can be learned by simple computational systems with no reference to numerosity. Our behavioural and computational results provide a counterexample against the assumption that 2D visual stimuli with different numbers of shapes are processed by honeybees as discrete numerical elements. Our findings suggest that an alternative non-numerical explanation exists for studies using similar methods in honeybees.”

• It would be important that the authors should provide the results of the two groups (bees trained on more-than and bees trained on less-than) separately, and that the effects at test of the different type of training are tested statistically in a separate way.

We apologise for the lack of clarity and details in our previous version. We now provide details on the glmms performed. We included rule (more-than/less-than) within a glmm and found it did not affect bee performance and therefore presented data within the figures as mean \pm s.e.m. of all bees. We clarify this on line 143 where we now state “For

the glmm evaluating the results of the tests, country and rule (more-than/less-than) were considered as fixed factors and bee ID as a random effect (Table S1). Because country and rule had no effect on performance, we display data as the mean \pm s.e.m. of all bees' data. We then removed country and rule in a second glmm (Table 2). Our second model ranked better than the first on the grounds of Akaike's Information Criterion [59] adjusted for small sample sizes (AICc), and therefore we present data from this second model in the main text." Further, in the legend of Figure 2 we now state "Data shown are combined from the two groups trained with different numerical rules since no difference in performance was found between groups (Table 1; Methods)."

We also now provide in the legend of Figure 2 the performance mean \pm sem for both rule groups, stating "In the Learning test, honeybees more often chose stimuli following the numerical rule on which they had been trained (71.3 \pm 3.3%; more-than: 70.3 \pm 4.7%; less-than: 72.4 \pm 4.8%). However, when tested on stimuli that differed in continuous cues but not number of elements (Equal/Incongruent test; middle bar; 32.5 \pm 2.6; more-than: 30.7 \pm 4.2%; less-than: 34.2 \pm 3.4%) and separately on two pairs of stimuli where numerosity and continuous cues were set in opposition (Incongruent/Opposite test; right bar; 36.7 \pm 1.8; more-than: 35.1 \pm 2.4%; less-than: 38.2 \pm 2.8%), honeybees chose stimuli based on continuous cues over numerosity." We also separated the individual bee's data points in Figure 2 into the two learning groups (empty and filled circles).

• It is not clear what kind of measurement the authors considered as dependent variable during the test phase, neither it is reported how many choices were scored during the test and the duration of this phase. Since previous studies on numerical abilities in honeybees used slightly different test phases, with either the consideration of a fixed amount of time during which each interaction with the stimuli is considered or the scoring of a fixed number of choices, the authors should report accurately these details of the test phase.

Thank you for pointing out this omission of details. On line 124 we now state "Each test lasts two minutes and all choices were recorded as the dependent variable for statistical analyses."

Other Comments:

• The author might consider reporting the average number of training trials complete by the subjects to reach the 80% of accuracy, as well as mean \pm s.e.m of the test performances for a better understanding of the graph 2b.

On lines 123 we now state "Bees reached criterion on an average of 41 \pm 8 choices." We also state now in the legend of Figure 2 the mean \pm s.e.m for each test.

• Line 58-61: it seems that some reference should be added to support the statement "[...]these results, along with other works suggesting honeybees and other animals are able to solve tasks in an unexpected ways[...]"

On line 53 we now provide references 2-7 in support of this statement.

2. Guiraud M, Roper M, Chittka L. 2018 High-speed videography reveals how honeybees can turn a spatial concept learning task into a simple discrimination task by stereotyped flight movements and sequential inspection of pattern elements. *Front. Psychol.* 9, 1347. (doi:10.3389/fpsyg.2018.01347)
3. Izquierdo A, Belcher AM. 2012 Rodent Models of Adaptive Decision Making. *Methods Mol. Biol. Clifton NJ* 829, 85–101. (doi:10.1007/978-1-61779-458-2_5)
4. Risko EF, Gilbert SJ. 2016 Cognitive Offloading. *Trends Cogn. Sci.* 20, 676–688. (doi:10.1016/j.tics.2016.07.002)
5. Jolicoeur P. 1988 Mental rotation and the identification of disoriented objects. *Can. J. Psychol.* 42, 461–478. (doi:10.1037/h0084200)
6. Wasserman EA, Zentall TR. 2006 *Comparative Cognition: Experimental Explorations of Animal Intelligence*. Oxford University Press.
7. Chittka L, Rossiter SJ, Skorupski P, Fernando C. 2012 What is comparable in comparative cognition? *Philos. Trans. R. Soc. Lond. B. Biol. Sci.* 367, 2677–2685. (doi:10.1098/rstb.2012.0215)

- **Line 296: the authors might consider change “contained” with “associated with”.**

To be clearer we have changed this sentence now on line 127 to “During all tests, 10 μ l of unrewarding water was placed on each platform.”

- **Line 297: the authors might consider change “reminder trials” with “refresh trials”, since the latter is more commonly used in studies on numerical abilities.**

Changed

- **Figure 2c-f: the authors should consider to enlarge those figures since it is quite difficult to appreciate the stimuli represented on the x axis.**

Enlarged

- **The author might consider to propose a specific name for each of the test presented (as for instance, learning test, continuous generalization test, continuous incongruent test) in order to help the reader also in the interpretation of the graphs.**

Thank you for this suggestion. We now call these “Learning test” where bees were tested with similar training stimuli but novel shapes, “Equal/Incongruent test” where stimuli contained the same number of elements, but differed in edge-length, convex hull, and spatial frequency, and “Incongruent/Opposite test” where stimuli differed in number and continuous cues in opposite directions.

- **The author wrote that all the continuous variables were tested simultaneously only once, in a study recently published (MaBouDi H, Dona HSG, Gatto E, Loukola OJ, Buckley E, Onoufriou PD, Skorupski P, Chittka L. 2020 Bumblebees use**

sequential scanning of countable items in visual patterns to solve numerosity tasks. *Integr. Comp. Biol.* (doi:10.1093/icb/icaa025)). I do not see how all continuous variables can be tested simultaneously, however. This is simply not possible. Looking at the paper I found that indeed the authors only presented stimuli with different element's dimension, shape and colour at test. Thus, several other continuous variables were not controlled.

We apologise for the lack of clarity and miscommunication. We now make clear that we mean "in one test". On lines 341 we now state "Testing all continuous variables (that cannot be kept constant across stimuli) and numerosity within one test can help determine if continuous variables have been learned.

We also provide clarity on this method by discussing the design in more detail on lines 343 where we now state "In one of our recent works, examining how bumblebees solved a numeric-based task, we assessed the use of continuous cues within one unrewarded test [79]. Here, bees were shown 10 stimuli simultaneously during one unrewarded test, each with different numbers of elements and levels of continuous cues. We chose the characteristics of different stimuli so that the bees' choices for some over others would reveal whether or not they had learned and used specific continuous cues to solve the task. For example, two displays both contained the same number of elements, but the elements in one of the displays had a greater edge-length. Bees chose these two displays equally in the test, suggesting they did not use edge length. However, if they had performed well on the test (i.e. more often chose stimuli based on the numerosity rule they had been trained) but had chosen one of these two stimuli significantly more than the other, this would suggest bees had learned and used edge-length instead of numerosity. We provided pairs of stimuli that varied in this way for edge-length, area, convex hull, spatial frequency and illusionary contour (Area was kept constant throughout training and tests and therefore did not need to be tested)."

• **Line 45: I think this should be substantiated by at least some general reference; in particular, as to the 'innate' part I believe the only direct evidence for that comes from studies in newborn chicks (see e.g. for a review: Vallortigara, G. (2017). An animal's sense of number. In "The nature and Development of Mathematics. Cross Disciplinary Perspective on Cognition, Learning and Culture" (Adams, J.W., Barmby P., Mesoudi, A., eds.), pp. 43-65, Routledge, New York.**

In an attempt to address concerns from both referees, and to improve the clarity of the manuscript, we have removed and replaced these sentences. However, we have added this reference to line 56 where we now state "Numerical cognition has been claimed in a large number of animal species (e.g. [8–39]), suggesting that a sense of number is widespread (for reviews see [40–42])."

• **Line 45: "Recent work" looks a bit weird, for the idea that numerosness encoding is based on magnitude dates back to classical work by Randy Gallistel and, after that, to the so-called ATOM Theory by Walsh. I think the authors should make an effort to provide a proper theoretical framework for the important issues they raised. In fact, even with insects there has been recent work arguing for**

generalization between the domains of discrete and continuous magnitudes that looks quite relevant to this paper (see Bortot et al (2020). Transfer from number to size reveals abstract coding of magnitude in honeybees. *iScience* 23, 101122 <https://doi.org/10.1016/j.isci.2020.101122>).

In an attempt to address concerns from both referees, and to improve the clarity of the manuscript, we have removed and replaced these sentences. We now state on line 56 “Numerical cognition has been claimed in a large number of animal species (e.g. [8–39]), suggesting that a sense of number is widespread (for reviews see [40–42]).” Our intent is only to point out that many animals have been shown to solve numerical cognition tasks. Given this, we feel, and hope you agree that providing theoretical background regarding number sense would be outside the scope of our current manuscript. We hope that you agree that by providing the above references, readers interested in this background will easily be able to find and read more.

With regards to generalisation between discrete and continuous magnitudes, we address this in our response to your comment below regarding previous line 107.

• **Line 50: Consider quoting a recent general review on number sense in invertebrates (Bortot et al (2020). A sense of number in invertebrates. *Biochemical and Biophysical Research Communications*, <https://doi.org/10.1016/j.bbrc.2020.11.039>**

Now cited in Introduction, line 57

• **Line 70: Among the most convincing evidence with good control conditions consider however the following: Rugani et al 2010. Imprinted numbers: Newborn chicks' sensitivity to number vs. continuous extent of objects they have been reared with. *Developmental Science*, 13: 790-797; and in frogs: Stancher et al. (2015). Numerical discrimination by frogs (*Bombina orientalis*). *Animal Cognition*, 18: 219-229.**

We have now changed and moved the previous line 70 to lines 79 where we state and include the Rugani et al 2010 citation you suggested: “Most studies investigating numerical cognition attempt to control for at least one non-numerical cue. Several works have made valiant efforts to control for most continuous cues (e.g. [57,58]). However, we have found no studies that tested for all continuous variables.”

• **Line 107: This seems to me not a logical conclusion: given the previous training, bees may be simply generalizing among different magnitudes as in the Bortot et al paper mentioned above.**

The point of our paper, and supported by our results, is that there is not sufficient evidence that bees have ever learned number. Our results indicate that bees did learn continuous cues. We concur that it is possible that bees might be generalizing from the available magnitudes, but number information is not required for this. As now stated in

the beginning of the Discussion on line 316 “Our behavioural and computational results provide a counterexample against the assumption that 2D visual stimuli with different numbers of shapes are processed by honeybees as discrete numerical elements. Our findings suggest that an alternative non-numerical explanation exists for studies using similar methods in honeybees.” We hope you agree that our extensive revisions make this message clearer.

• **Para. 110-119: Here I think disentangling data for learning “more” or “less” would be crucial. Also, I am wondering whether a different explanation could be as follows. Let’s suppose that during training bees encode different dimensions of magnitude, both discrete (number) and continuous (edge length, convex hull, and spatial frequency). Given that at test the authors introduced massive changes in continuous variables (at least 3 dimensions vs. the only 1 of discrete) it may appear not surprising that bees tended to use continuous variables.**

We now make clear in our Materials and Methods that our glmm included rule (more-than/less-than) as a fixed factor and that rule has no effect on performance in any of the tests. We now state on line 143 “For the glmm evaluating the results of the tests, country and rule (more-than/less-than) were considered as fixed factors and bee ID as a random effect (Table S1). Because country and rule had no effect on performance, we display data as the mean \pm s.e.m. of all bees’ data. We then removed country and rule in a second glmm (Table 2). Our second model ranked better than the first on the grounds of Akaike’s Information Criterion [59] adjusted for small sample sizes (AICc), and therefore we present data from this second model in the main text.” We also provide in the legend of Figure 2 the performance mean \pm sem for both rule groups. We also separated the individual bee’s data points in Figure 2 into the two learning groups (empty and filled circles).

It may be that bees learned numerosity in this task, but there is not sufficient evidence to support such a claim. Our results indicate that bees did learn continuous cues and thus did not require numerosity to solve the task. Therefore, as we state on line 319 “an alternative non-numerical explanation exists for studies using similar methods in honeybees.”

• **Line 135: As I stated above this statement contrasts strikingly with what is reported in the Supplementary materials of the Science paper, in which spatial frequency seems indeed to have been controlled for.**

Please see our response to your previous comment above.

Appendix B

01/15/2021

Dear Dr Brosnan,

Please find attached a revised version of the manuscript **Non-numerical strategies used by bees to solve numerical cognition tasks** (RSPB-2020-2711.R1), which we would like to resubmit as a *Research Article to Proc B*.

On January 12, 2020, you had sent us an email stating our manuscript had been accepted pending minor revisions suggested by the reviewers. We have now made these revisions, which include:

- (1) We revised the abstract to make clearer the initial description of our model, as suggested by Referee #1.
- (2) In response to Referee #1, we replaced the few times we used the term “element” with “neuron” in our initial description of our neural network model and further explain to Referee #1 why only seven neurons can obtain the necessary spatial frequency information required to discriminate the stimuli.
- (3) In response to Referee # 2 regarding Bortot et al. 2020, we explain that
 - a. Leibovich T, Henik A. 2014 [REF#57] showed that the method used in Bortot et al. 2020 does not eliminate learning and use of continuous cues.
 - b. No analysis was done to determine if a correlation existed between number and continuous cues.
 - c. No analysis was done to determine if bees’ performances were affected by the subgroup of stimuli used in training or testing.
 - d. Because insufficient evidence is provided for the assumption that bees discriminated based on number, the results of the “size-generalization test” of Bortot et al. 2020 actually indicate that a continuous cue was learned and used to solve the task.

Below, we first repeat the referee comments in bold and then follow each with our answers in italics. Thank you for your time and effort.

Kind regards,

Cwyn Solvi

Referee #1

Overall I am satisfied with the changes made in response to my original review. I have the following minor changes to suggest.

Abstract - The first sentence of the abstract is rather awkwardly phrased, and seems overly general and not really necessary. The term 'numeric-based' in the second sentence is unclear, and the sense could be more clearly conveyed, e.g. by "We examined how bees solve a decision task that uses stimuli commonly found in numerical cognition studies". I also think the phrase "a simple network

model containing just nine elements" is not an adequate reflection of the model; it would be better to say something like "a model using biologically plausible spatial frequency filtering and a simple associative rule". 'Nine elements' is not meaningful when the complexity of each element is unknown.

We have deleted the first sentence of the abstract and changed the second sentence to "We examined how bees solve a visual discrimination task that uses stimuli commonly found in numerical cognition studies." We have also changed the sentence referring to the model to "A simple network model using biologically plausible visual feature filtering and a simple associative rule was capable of learning the task using only continuous cues inherent in the training stimuli, with no numerical processing."

I appreciate that the authors have now clarified why pre-processing using frequency filtering is more plausible than 'numeric' filtering, but there is still some difference between (local) Gabor-like filters and (global) Fourier analysis that mean the model is not as 'simple' as they repeatedly claim. E.g how many simple and complex cells might be needed, and their output integrated in what way (in subsequent layers?), to produce the same response as one 'element' in their model which is tuned to a preferred Fourier frequency for the whole image? It also seems unnecessary to describe the decision element as "a neuron in the mushroom bodies" given the abstraction level of this model.

We realise that in a few instances, we used the term element. To avoid confusion, we are now consistent throughout the manuscript and use neuron. Each neuron in our model represents one real neuron in the bee brain. Please note that we performed Fourier transformation, the classical method in image processing, to obtain the spatial frequencies of training and test stimuli and determine that spatial frequency correlated with numerosity (Methods; Fig 1j and 2f). In the brain, extracting spatial frequency information from stimuli does not require Fourier transformation, but rather can be obtained through Gabor filters. It is known that V1 neurons, acting as Gabor filters, in the early processing stages of the visual system of animals, extract different local spatial frequencies. Similar frequency sensitive neurons in the visual lobes of insects are large field, which means they likely encompass an entire training stimulus. In our model, each neuron (representing one real sensory neuron) is tuned to a specific spatial frequency. Therefore, each neuron will respond more to stimuli with a specific number of elements than any other neuron. Hence, seven sensory neurons are enough to discriminate between stimuli with 0-6 items based on their different spatial frequencies.

We have now removed mention of the decision neuron being located in the mushroom bodies.

Referee #2

I believe the authors have addressed adequately all my concerns, and that the paper deserves to be published. I have only one final issue.

On p. 3 lines 81-82 the Authors stated that they found no studies that tested for all continuous variables. It seems to me, however, that the paper by Bortot et al

(2020) Transfer from number to size reveals abstract coding of magnitude in honeybees. *iScience* 23, 101122 <https://doi.org/10.1016/j.isci.2020.101122>, that they did not cite, is in fact providing an example, of such a control for the authors of this paper checked for overall area, perimeter (contour length), convex hull and density. They also balanced the presence of the largest element (a third of the time it was in the smaller number group, a third of the time in the larger number group and a third of the time was present in both).

The only parameter they did not check for during training was spatial frequency, however, if the bees were using this parameter one should have expected to observe that in the space (size) generalization test: i.e., that the bees trained to choose the largest number, which could contain overall the relatively smallest dots, would have had to choose the smaller elements at test, and vice versa those trained to choose the smaller number that could contain overall the relatively largest elements in half of the cases, they should have chosen the largest elements at test. The opposite was observed. Thus, as far as I can judge, this paper does in fact provide a control for all continuous variables.

Bortot et al. 2020 (iScience 23, 101122), now referenced in the new version of our manuscript, did not control for all continuous variables nor do their methods eliminate the possible use of continuous cues. Each continuous variable was separately kept constant and only in a subset (one quarter to one half) of all stimuli. As pointed out on line 328, Leibovich T, Henik A. 2014 [REF#57] showed that this method does not eliminate use of continuous cues. Bortot et al. 2020 also provided no analysis to show that continuous variables do not covary with number across the stimulus set. Further, no analysis was done on the bees' performance on the different subsets of stimuli, which might provide information on which cues were being learned/used to solve the task. Like many other numerical cognition studies, Bortot et al. 2020 provide insufficient evidence for the assumption that the animals discriminated stimuli based on number. Therefore, the results of the "size generalization test" suggest that bees used continuous variables rather than numerosity.